# ADAPT AND DIFFUSE: SAMPLE-ADAPTIVE RECONSTRUCTION VIA LATENT DIFFUSION MODELS

## ABSTRACT

Inverse problems arise in a multitude of applications, where the goal is to recover a clean signal from noisy and possibly (non)linear observations. The difficulty of a reconstruction problem depends on multiple factors, such as the structure of the ground truth signal, the severity of the degradation, the implicit bias of the reconstruction model and the complex interactions between the above factors. This results in natural sample-by-sample variation in the difficulty of a reconstruction task, which is often overlooked by contemporary techniques. Recently, diffusion-based inverse problem solvers have established new state-of-the-art in various reconstruction tasks. Our key observation in this paper is that most existing solvers lack the ability to adapt their compute power to the difficulty of the reconstruction task, resulting in long inference times, subpar performance and wasteful resource allocation. We propose a novel method that we call severity encoding, to estimate the degradation severity of noisy, degraded signals in the latent space of an autoencoder. We show that the estimated severity has strong correlation with the true corruption level and can give useful hints at the difficulty of reconstruction problems on a sample-by-sample basis. Furthermore, we propose a reconstruction method based on latent diffusion models that leverages the predicted degradation severities to fine-tune the reverse diffusion sampling trajectory and thus achieve sample-adaptive inference times. We perform numerical experiments on both linear and nonlinear inverse problems and demonstrate that our technique achieves performance comparable to state-of-the-art diffusion-based techniques, with significant improvements in computational efficiency.

## 1    INTRODUCTION

Inverse problems arise in a multitude of computer vision Ledig et al. (2017); Kupyn et al. (2018); Wang et al. (2018), biomedical imaging Ardila et al. (2019); Sriram et al. (2020) and scientific Hand et al. (2018); Feng et al. (2023) applications, where the goal is to recover a clean signal from noisy and degraded observations. As information is fundamentally lost in the process, structural assumptions on the clean signal are needed to make recovery possible. Traditional compressed sensing Candes et al. (2006); Donoho (2006) approaches utilize explicit regularizers that encourage sparsity of the reconstruction in transformation domains such as wavelets. More recently, data-driven supervised and unsupervised deep learning methods have established new state-of-the-art in tackling most inverse problems (see an overview in Ongie et al. (2020)).

A key shortcoming of available techniques is their inherent inability to adapt their compute power allocation to the difficulty of reconstructing a given corrupted sample. There is a natural sample-by-sample variation in the difficulty of recovery due to multiple factors. First, variations in the measurement process (e. g. more or less additive noise, different blur kernels) greatly impact the difficulty of reconstruction. Second, a sample can be inherently difficult to reconstruct for the particular model, if it is different from examples seen in the training set (out-of-distribution samples). Third, the amount of information loss due to the interaction between the specific sample and the applied degradation can vary vastly. For instance, applying a blur kernel to an image consisting of high-frequency textures destroys significantly more information than applying the same kernel to a smooth image. Finally, the implicit bias of the model architecture towards certain signal classes (e.g. piece-wise constant or smooth for convolutional architectures) can be a key factor in determining the

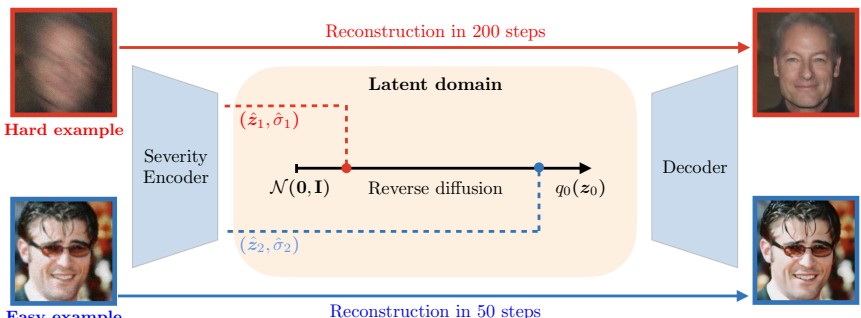

Figure 1: Overview of our method: we estimate the degradation severity of corrupted images in the latent space of an autoencoder. We leverage the severity predictions to find the optimal start time in a latent reverse diffusion process on a sample-by-sample basis. As a result, inference cost is automatically scaled by the difficulty of the reconstruction task at test time.

difficulty of a recovery task. Therefore, expending the same amount of compute to reconstruct all examples is potentially wasteful, especially on datasets with varied corruption parameters.

Sample-adaptive methods that incorporate the difficulty of a reconstruction problem, or *the severity of degradation*, and allocate compute effort accordingly are thus highly desired. To the best of our knowledge, such methods have not been studied extensively in the literature. *Unrolled networks* Zhang & Ghanem (2018); Sun et al. (2016) have been proposed for reconstruction, that map the iterations of popular optimizers to learnable submodules, where deeper networks can be used to tackle more challenging reconstruction tasks. However, network size is determined in training time and therefore these methods are unable to adapt on a sample-by-sample basis. Deep Equilibrium Models Bai et al. (2019) have been proposed to solve inverse problems Gilton et al. (2021) by training networks of arbitrary depth through the construction of fixed-point iterations. These methods can adapt their compute in test time by scaling the number of fixed-point iterations to convergence, however it is unclear how the optimal number of iterations correlates with degradation severity.

Diffusion models have established new state-of-the-art performance in synthesizing data of various modalities Dhariwal & Nichol (2021); Nichol et al. (2021); Ramesh et al. (2022); Rombach et al. (2022); Saharia et al. (2022b); Ho et al. (2022a); Saharia et al. (2022a); Ho et al. (2022b); Kong et al. (2020), inverse problem solving and image restoration Kadkhodaie & Simoncelli (2021); Saharia et al. (2021); Song et al. (2021b); Chung & Ye (2022); Chung et al. (2022c;a;b); Kawar et al. (2021; 2022a;b); Fabian et al. (2023). Diffusion-based sampling techniques generate the missing information destroyed by the corruption step-by-step through a diffusion process that transforms pure noise into a target distribution. Recent work Chung et al. (2022c) has shown that the sampling trajectory can be significantly shortened by starting the reverse diffusion process from a good initial reconstruction, instead of pure noise. However, this approach treats the noise level of the starting manifold as a hyperparameter independent of degradation severity. Therefore, even though sampling is accelerated, the same number of function evaluations are required to reconstruct any sample. In Fabian et al. (2023) an early stopping technique is proposed for diffusion-based reconstruction, however it is unclear how to determine the stopping time on a sample-by-sample basis.

More recently, latent domain diffusion, that is a diffusion process defined in the low-dimensional latent space of a pre-trained autoencoder, has demonstrated great success in image synthesis Rombach et al. (2022) and has been successfully applied to solving linear inverse problems Rout et al. (2023) and in high-resolution image restoration Luo et al. (2023). Latent diffusion has the clear benefit of improved efficiency due to the reduced dimensionality of the problem leading to faster sampling. In addition to this, the latent space consists of compressed representations of relevant information in data and thus provides a natural space to quantify the loss of information due to image corruptions, which strongly correlates with the difficulty of the reconstruction task.

In this paper, we propose a novel reconstruction framework (Figure 1), where the cost of inference is automatically scaled based on the difficulty of the reconstruction task on a sample-by-sample basis. Our contributions are as follows:

- We propose a novel method that we call severity encoding, to estimate the degradation severity of noisy, degraded images in the latent space of an autoencoder. We show that the estimated severity has strong correlation with the true corruption level and can give useful hints at the difficulty of reconstruction problems on a sample-by-sample basis. Training the severity encoder is efficient, as it can be done by fine-tuning a pre-trained encoder.

- We propose a reconstruction method based on latent diffusion models that leverages the predicted degradation severities to fine-tune the reverse diffusion sampling trajectory and thus achieve sample-adaptive inference times. Furthermore, we utilize latent diffusion posterior sampling to maintain data consistency with the observations. Our framework can take advantage of pre-trained latent diffusion models out of the box, reducing compute requirements compared to other image-domain diffusion solvers. We call our method Flash-Diffusion: Fast Latent Sample-Adaptive Reconstruction ScHeme.

- We perform numerical experiments on both linear and nonlinear inverse problems and demonstrate that the proposed technique achieves performance comparable to state-of-the-art diffusion-based techniques, while significantly reducing the computational cost.

## 2 BACKGROUND

**Diffusion models –** Diffusion in the context of generative modeling consists of transforming a clean data distribution $x_0 \sim q_0(x)$ through a forward noising process, defined over $0 \leq t \leq T$, $t \in \mathbb{R}$, into some tractable distribution $q_T$. Typically, $q_t$ is chosen such that $x_t$ is obtained from $x_0$ via adding *i.i.d.* Gaussian noise, that is $q_t(x_t|x_0) \sim \mathcal{N}(x_0, \sigma_t^2 \mathbf{I})$, where $\sigma_t^2$ is from a known variance schedule. Diffusion models (DMs) Sohl-Dickstein et al. (2015); Ho et al. (2020); Song & Ermon (2020a;b) learn to reverse the forward corruption process in order to generate data samples starting from a simple Gaussian distribution. The forward process can be described as an Itô stochastic differential equation (SDE) Song et al. (2020) $dx = f(x, t)dt + g(t)dw$, where $f$ and $g$ are also called the *drift* and *diffusion* coefficients, and $w \in \mathbb{R}^n$ is the standard Wiener process. The forward SDE is reversible Anderson (1982) and the reverse SDE can be written as

$$dx = [f(x, t) - g(t)^2 \nabla_x \log q_t(x)]dt + g(t)d\bar{w}, \quad (1)$$

where $\bar{w}$ is the standard Wiener process where time flows in the negative direction and $\nabla_x \log q_t(x)$ is referred to as the *score* of the data distribution. The score is approximated by a neural network $s_\theta(x_t, t)$ trained such that $s_\theta(x, t) \approx \nabla_x \log q_t(x)$. Then, $s_\theta(x_t, t)$ can be used to simulate the reverse SDE in (1) from which a variety of discrete time sampling algorithms can be derived. The continuous time interval $t \in [0, T]$ is typically discretized uniformly into $N$ time steps.

Denoising Diffusion Probabilistic Models (DDPMs) Sohl-Dickstein et al. (2015); Ho et al. (2020) are obtained from the discretization of the variance preserving SDE with $f(x, t) = -\frac{1}{2}\beta_t x$ and $g(t) = \sqrt{\beta_t}$, where $\beta_t$ is a pre-defined variance schedule that is a strictly increasing function of $t$. One can sample from the corresponding forward diffusion process at any time step $i$ as $x_i = \sqrt{\bar{\alpha}_i} x_0 + \sqrt{1 - \bar{\alpha}_i}\varepsilon$, with $\varepsilon \sim \mathcal{N}(\mathbf{0}, \mathbf{I})$ and $\bar{\alpha}_i = \prod_{j=1}^{i} \alpha_i$, $\alpha_i = 1 - \beta_i$. By minimizing the denoising score-matching objective

$$L_{DM} = \mathbb{E}_{\varepsilon \sim \mathcal{N}(\mathbf{0}, \mathbf{I}), i \sim \mathcal{U}[1, N], x_i \sim q_0(x_0)q_i(x_i|x_0)} \left[ \|\varepsilon_\theta(x_i, i) - \varepsilon\|^2 \right]$$

the (rescaled) score model $\varepsilon_\theta(x_i, i) = -\sqrt{1 - \bar{\alpha}_i} s_\theta(x_i, i)$ learns to predict the noise on the input corrupted signal. The associated reverse diffusion step derived from (1) takes the form

$$x_{i-1} = \frac{1}{\sqrt{\alpha_i}} (x_i + (1 - \alpha_i)s_\theta(x_i, i)) + \sqrt{1 - \alpha_i}\varepsilon,$$

which is iterated from $i = N$ to $i = 1$ to draw a sample from the data distribution, starting from $x_N \sim \mathcal{N}(\mathbf{0}, \mathbf{I})$.

**Latent Diffusion Models (LDMs) –** LDMs Rombach et al. (2022) aim to mitigate the computational burden of traditional diffusion models by running diffusion in a low-dimensional latent space of a pre-trained autoencoder. In particular, an encoder $\mathcal{E}_0$ is trained to extract a compressed representation $z \in \mathbb{R}^d$, $d << n$ of the input signal $x$ in the form $z = \mathcal{E}_0(x)$. To recover the clean signal from the latent representation $z$, a decoder $\mathcal{D}_0$ is trained such that $\mathcal{D}_0(\mathcal{E}_0(x)) \approx x$. A score model that progressively denoises $z$ can be trained in the latent space of the autoencoder via the objective

$$L_{LDM} = \mathbb{E}_{\varepsilon \sim \mathcal{N}(\mathbf{0},\mathbf{I}), i \sim \mathcal{U}[1,N], \boldsymbol{z}_i \sim \tilde{q}_0(\boldsymbol{z}_0)\tilde{q}_i(\boldsymbol{z}_i|\boldsymbol{z}_0)} \left[ \|\varepsilon_{\boldsymbol{\theta}}(\boldsymbol{z}_i, i) - \varepsilon\|^2 \right],$$

where $\boldsymbol{z}_0 = \mathcal{E}_0(\boldsymbol{x}_0)$, $\boldsymbol{x}_0 \sim q_0(\boldsymbol{x}_0)$ and $\tilde{q}_i(\boldsymbol{z}_i|\boldsymbol{z}_0) \sim \mathcal{N}(\sqrt{\bar{\alpha}_i}\boldsymbol{z}_0, (1 - \bar{\alpha}_i)\mathbf{I})$ following the DDPM framework. The final generated image can be obtained by passing the denoised latent through $\mathcal{D}_0$.

**Diffusion models for solving inverse problems –** Solving a general noisy inverse problem amounts to finding the clean signal $\boldsymbol{x} \in \mathbb{R}^n$ from a noisy and degraded observation $\boldsymbol{y} \in \mathbb{R}^m$ in the form

$$\boldsymbol{y} = \mathcal{A}(\boldsymbol{x}) + \boldsymbol{n}, \tag{2}$$

where $\mathcal{A} : \mathbb{R}^n \to \mathbb{R}^m$ denotes a deterministic degradation (such as blurring or inpainting) and $\boldsymbol{n} \sim \mathcal{N}(\mathbf{0}, \sigma_y^2\mathbf{I})$ is *i.i.d.* additive Gaussian noise. As information is fundamentally lost in the measurement process, structural assumptions on clean signals are necessary to recover $\boldsymbol{x}$. Deep learning approaches extract a representation of clean signals from training data either by learning to directly map $\boldsymbol{y}$ to $\boldsymbol{x}$ or by learning a generative model $p_\theta(\boldsymbol{x})$ that represents the underlying structure of clean data and can be leveraged as a prior to solve (2). In particular, the posterior over clean data can be written as $p_\theta(\boldsymbol{x}|\boldsymbol{y}) \propto p_\theta(\boldsymbol{x})p(\boldsymbol{y}|\boldsymbol{x})$, where the likelihood $p(\boldsymbol{y}|\boldsymbol{x})$ is represented by (2). Thus, one can sample from the posterior by querying the generative model. The score of the posterior log-probabilities can be written as $\nabla_{\boldsymbol{x}} \log p_\theta(\boldsymbol{x}|\boldsymbol{y}) = \nabla_{\boldsymbol{x}} \log p_\theta(\boldsymbol{x}) + \nabla_{\boldsymbol{x}} \log p(\boldsymbol{y}|\boldsymbol{x})$, where the first term corresponds to an unconditional score model trained to predict noise on the signal without any information about the forward model $\mathcal{A}$. The score of the likelihood term however is challenging to estimate in general. Various approaches have emerged to incorporate the data acquisition model into an unconditional diffusion process, including projection-based approaches Song et al. (2021b); Chung & Ye (2022); Chung et al. (2022c), restricting updates to stay on a given manifold Chung et al. (2022a;b), spectral approaches Kawar et al. (2022a), or methods that tailor the diffusion process to the degradation Welker et al. (2022); Fabian et al. (2023); Delbracio & Milanfar (2023).

A key challenge of diffusion-based solvers is their heavy compute demand, as reconstructing a single sample requires typically $100 - 1000$ evaluations of a large score model. Come-Closer-Diffuse-Faster (CCDF) Chung et al. (2022c), a recently proposed solver shortens the sampling trajectory by leveraging a good initial posterior mean estimate $\hat{\boldsymbol{x}}_0$ from a reconstruction network. They initialize the reverse process by jumping to a fixed time step in the forward process via $\boldsymbol{x}_k = \sqrt{\bar{\alpha}_k}\hat{\boldsymbol{x}}_0 + \sqrt{1 - \bar{\alpha}_k}\varepsilon$, and only perform $k << N$ reverse diffusion steps, where $k$ is a fixed hyperparameter.

## 3 METHOD

### 3.1 SEVERITY ENCODING

The goal of inverse problem solving is to recover the clean signal $\boldsymbol{x}$ from a corrupted observation $\boldsymbol{y}$ (see (2)). The degradation $\mathcal{A}$ and additive noise $\boldsymbol{n}$ fundamentally destroy information in $\boldsymbol{x}$. The amount of information loss, or the *severity* of the degradation, strongly depends on the interaction between the signal structure and the specific degradation. For instance, blurring removes high-frequency information, which implies that applying a blur kernel to an image with abundant high-frequency detail (textures, hair, background clutter etc.) results in a *more severe degradation* compared to applying the same kernel to a smooth image with few details. In other words, the difficulty of recovering the clean signal does not solely depend on the degradation process itself, but also on the specific signal the degradation is applied to. Thus, tuning the reconstruction method's capacity purely based on the forward model misses a key component of the problem: the data itself.

Quantifying the severity of a degradation is a challenging task in image domain. As an example, consider the forward model $\boldsymbol{y} = c\boldsymbol{x}$, $c \in \mathbb{R}^+$ that simply rescales the clean signal. Recovery of $\boldsymbol{x}$ from $\boldsymbol{y}$ is trivial, however image similarity metrics such as PSNR or NMSE that are based on the Euclidean distance in image domain may indicate arbitrarily large discrepancy between the degraded and clean signals. On the other hand, consider $\boldsymbol{y} = \boldsymbol{x} + \boldsymbol{n}$, $\boldsymbol{n} \sim \mathcal{N}(\mathbf{0}, \sigma^2\mathbf{I})$ where the clean signal is simply perturbed by some additive random noise. Even though the image domain perturbation is (potentially) small, information is fundamentally lost and perfect reconstruction is no longer possible.

What is often referred to as the *manifold hypothesis* Bengio et al. (2013) states that natural images live in a lower dimensional manifold embedded in $n$-dimensional pixel-space. This in turn implies that the information contained in an image can be represented by a low-dimensional latent vector that encapsulates the relevant features of the image. Autoencoders Kingma & Welling (2013); Razavi et al. (2019) learn a latent representation from data by first summarizing the input image into a compressed latent vector $\boldsymbol{z} = \mathcal{E}_0(\boldsymbol{x})$ through an encoder. Then, the original image can be recovered from the

latent via the decoder $\hat{\boldsymbol{x}} = \mathcal{D}_0(\boldsymbol{z})$ such that $\boldsymbol{x} \approx \hat{\boldsymbol{x}}$. As the latent space of autoencoders contains only the relevant information of data, it is a more natural space to quantify the loss of information due to the degradation than the image domain.

In particular, assume that we have access to the latent representation of clean images $\boldsymbol{z}_0 = \mathcal{E}_0(\boldsymbol{x}_0)$, $\boldsymbol{z}_0 \in \mathbb{R}^d$, for instance from a pre-trained autoencoder. We propose a *severity encoder* $\hat{\mathcal{E}}_{\boldsymbol{\theta}}$ that achieves two objectives simultaneously: (1) it can predict the latent representation of a clean image, given a noisy and degraded observation and (2) it can quantify the error in its own latent estimation. We denote $\hat{\mathcal{E}}_{\boldsymbol{\theta}}(\boldsymbol{y}) = (\hat{\boldsymbol{z}}, \hat{\sigma})$ with $\hat{\boldsymbol{z}} \in \mathbb{R}^d$ the estimate of $\boldsymbol{z}_0$ and $\hat{\sigma} \in \mathbb{R}$ the estimated degradation severity to be specified shortly. We use the notation $\hat{\mathcal{E}}_{\boldsymbol{z}}(\boldsymbol{y}) = \hat{\boldsymbol{z}}$ and $\hat{\mathcal{E}}_{\sigma}(\boldsymbol{y}) = \hat{\sigma}$ for the two conceptual components of our model, however in practice a single architecture is used to represent $\hat{\mathcal{E}}_{\boldsymbol{\theta}}$. The first objective can be interpreted as image reconstruction in the latent space of the autoencoder: for $\boldsymbol{y} = \mathcal{A}(\boldsymbol{x}) + \boldsymbol{n}$ and $\boldsymbol{z}_0 = \mathcal{E}_0(\boldsymbol{x})$, we have $\hat{\mathcal{E}}_{\boldsymbol{z}}(\boldsymbol{y}) = \hat{\boldsymbol{z}} \approx \boldsymbol{z}_0$. The second objective captures the intuition that recovering $\boldsymbol{z}_0$ from $\boldsymbol{y}$ exactly may not be possible, and the prediction error is proportional to the loss of information about $\boldsymbol{x}$ due to the corruption. Thus, even though the predicted latent $\hat{\boldsymbol{z}}$ might be away from the true $\boldsymbol{z}_0$, the encoder quantifies the uncertainty in its own prediction. More specifically, we make the assumption that the prediction error in latent space can be modeled as zero-mean *i.i.d.* Gaussian. That is, $\boldsymbol{e}(\boldsymbol{y}) = \hat{\boldsymbol{z}} - \boldsymbol{z}_0 \sim \mathcal{N}(\boldsymbol{0}, \sigma_*^2(\boldsymbol{y})\mathbf{I})$ and we interpret the variance in prediction error $\sigma_*^2$ as the measure of degradation severity. We optimize the joint objective

$$\mathbb{E}_{\boldsymbol{x}_0 \sim q_0(\boldsymbol{x}_0), \boldsymbol{y} \sim \mathcal{N}(\mathcal{A}(\boldsymbol{x}_0), \sigma_y^2 \mathbf{I})} \left[ \left\| \boldsymbol{z}_0 - \hat{\mathcal{E}}_{\boldsymbol{z}}(\boldsymbol{y}) \right\|^2 + \lambda_\sigma \left\| \bar{\sigma}^2(\boldsymbol{y}, \boldsymbol{z}_0) - \hat{\mathcal{E}}_\sigma(\boldsymbol{y}) \right\|^2 \right] := L_{lat.rec.} + \lambda_\sigma L_{err.}, \quad (3)$$

with $\boldsymbol{z}_0 = \mathcal{E}_0(\boldsymbol{x}_0)$ for a fixed, pre-trained encoder $\mathcal{E}_0$ and $\bar{\sigma}^2(\boldsymbol{y}, \boldsymbol{z}_0) = \frac{1}{d-1} \sum_{i=1}^d (\boldsymbol{e}^{(i)} - \frac{1}{d} \sum_{j=1}^d \boldsymbol{e}^{(j)})^2$ is the sample variance of the prediction error estimating $\sigma_*^2$. Here $\lambda_\sigma > 0$ is a hyperparameter that balances between reconstruction accuracy ($L_{lat.rec.}$) and error prediction performance ($L_{err.}$). We empirically observe that even small loss values of $L_{lat.rec.}$ (that is fairly good latent reconstruction) may correspond to visible reconstruction error in image domain as semantically less meaningful features in image domain are often not captured in the latent representation. Therefore, we utilize an extra loss term that imposes image domain consistency with the ground truth image in the form

$$L_{im.rec.} = \mathbb{E}_{\boldsymbol{x}_0 \sim q_0(\boldsymbol{x}_0), \boldsymbol{y} \sim \mathcal{N}(\mathcal{A}(\boldsymbol{x}_0), \sigma_y^2 \mathbf{I})} \left[ \left\| \boldsymbol{x}_0 - \mathcal{D}_0(\hat{\mathcal{E}}_{\boldsymbol{z}}(\boldsymbol{y})) \right\|^2 \right],$$

resulting in the final combined loss $L_{sev} = L_{lat.rec.} + \lambda_\sigma L_{err.} + \lambda_{im.} L_{im.rec.}$, with $\lambda_{im.} \geq 0$ hyperparameter. Training the severity encoder is fast, as one can fine-tune the pre-trained encoder $\mathcal{E}_0$.

## 3.2 Sample-adaptive Inference

Diffusion-based inverse problem solvers synthesize missing data that has been destroyed by the degradation process through diffusion. As shown in Figure 2, depending on the amount of missing information (easy vs. hard samples), the optimal number of diffusion steps may greatly vary. Too few steps may not allow the diffusion process to generate realistic details on the image, leaving the reconstruction overly smooth. On the other hand, diffusion-based solvers are known to hallucinate details that may be inconsistent with the ground truth signal, or even become unstable when too many reverse diffusion steps are applied. Authors in Chung et al. (2022c) observe that there always exists an optimal spot between $0$ and $N$ diffusion steps that achieves the best reconstruction performance. We aim to automatically find this "sweet spot" on a sample-by-sample basis.

Our proposed severity encoder learns to map degraded signals to a noisy latent representation, where the noise level is proportional to the degradation severity. This provides us the unique opportunity to leverage a latent diffusion process to progressively denoise the latent estimate from our encoder. Even more importantly, we can automatically scale the number of reverse diffusion steps required to reach the clean latent manifold based on the predicted degradation severity.

**Finding the optimal starting time –** We find the time index $i_{start}$ in the latent diffusion process at which the signal-to-noise ratio (SNR) matches the SNR predicted by the severity encoder. Assume that the latent diffusion process is specified by the conditional distribution $q_i(\boldsymbol{z}_i | \boldsymbol{z}_0) \sim \mathcal{N}(a_i \boldsymbol{z}_0, b_i^2 \mathbf{I})$, where $a_i$ and $b_i$ are determined by the specific sampling method (e. g. $a_i = \sqrt{\bar{\alpha}_i}$ and $b_i^2 = 1 - \bar{\alpha}_i$ for DDPM). On the other hand, we have the noisy latent estimate $\hat{\boldsymbol{z}} \sim \mathcal{N}(\boldsymbol{z}_0, \sigma_*^2(\boldsymbol{y})\mathbf{I})$, where we

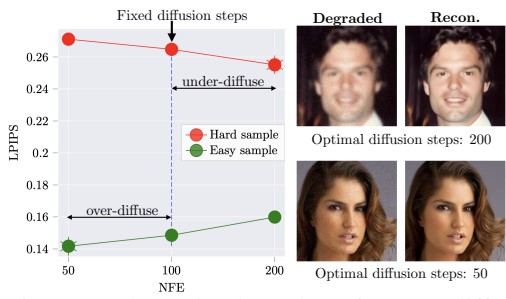
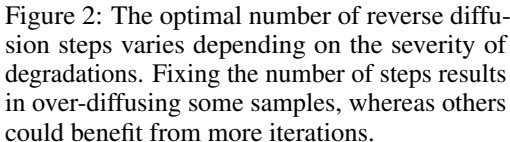
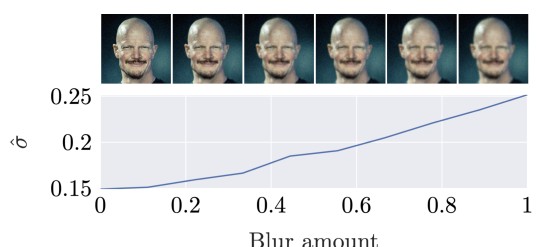

Figure 2: The optimal number of reverse diffusion steps varies depending on the severity of degradations. Fixing the number of steps results in over-diffusing some samples, whereas others could benefit from more iterations.

Figure 3: Effect of degradation on predicted severities: given a ground truth image corrupted by varying amount of blur, $\hat{\sigma}$ is a non-decreasing function of the blur amount.

estimate $\sigma_*^2$ by $\hat{\mathcal{E}}_\sigma(\boldsymbol{y})$. Then, SNR matching gives us the starting time index

$$i_{start}(\boldsymbol{y}) = \arg \min_{i \in [1,2,..,N]} \left| \frac{a_i^2}{b_i^2} - \frac{1}{\hat{\mathcal{E}}_\sigma(\boldsymbol{y})} \right| \tag{4}$$

Thus, we start reverse diffusion from the initial reconstruction $\hat{z}$ provided by the severity encoder and progressively denoise it using a pre-trained unconditional score model, where the length of the sampling trajectory is directly determined by the predicted severity of the degraded example.

**Noise correction –** Even though we assume that the prediction error in latent space is *i.i.d.* Gaussian in order to quantify the estimation error by a single scalar, in practice the error often has some structure. This can pose a challenge for the score model, as it has been trained to remove isotropic Gaussian noise. We observe that it is beneficial to mix $\hat{z}$ with some *i.i.d. correction noise* in order to suppress structure in the prediction error. In particular, we initialize the reverse process by

$$\boldsymbol{z}_{start} = \sqrt{1 - c\hat{\sigma}^2}\hat{z} + \sqrt{c\hat{\sigma}^2}\boldsymbol{\varepsilon}, \ \ \boldsymbol{\varepsilon} \sim \mathcal{N}(\boldsymbol{0}, \mathbf{I})$$

where $c \geq 0$ is a tuning parameter.

**Latent Diffusion Posterior Sampling –** Maintaining consistency with the measurements is non-trivial in the latent domain, as common projection-based approaches are not applicable directly in latent space. We propose Latent Diffusion Posterior Sampling (LDPS), a variant of diffusion posterior sampling Chung et al. (2022a) that guides the latent diffusion process towards data consistency in the original data space. In particular, by applying Bayes' rule the score of the posterior in latent space can be written as

$$\nabla_{\boldsymbol{z}_t} \log q_t(\boldsymbol{z}_t|\boldsymbol{y}) = \nabla_{\boldsymbol{z}_t} \log q_t(\boldsymbol{z}_t) + \nabla_{\boldsymbol{z}_t} \log q_t(\boldsymbol{y}|\boldsymbol{z}_t).$$

The first term on the r.h.s. is simply the unconditional score that we already have access to as pre-trained LDMs. As $q_t(\boldsymbol{y}|\boldsymbol{z}_t)$ cannot be written in closed form, following DPS we use the approximation $\nabla_{\boldsymbol{z}_t} \log q_t(\boldsymbol{y}|\boldsymbol{z}_t) \approx \nabla_{\boldsymbol{z}_t} \log q_t(\boldsymbol{y}|\hat{z}_0(\boldsymbol{z}_t))$, where $\hat{z}_0(\boldsymbol{z}_t) = \mathbb{E}[\boldsymbol{z}_0|\boldsymbol{z}_t]$ is the posterior mean of $\boldsymbol{z}_0$, which is straightforward to estimate from the score model via Tweedie's formula. This form is similar to PSLD in Rout et al. (2023), but without the "gluing" objective. As $\boldsymbol{y} \sim \mathcal{N}(\mathcal{A}(\mathcal{D}_0(\boldsymbol{z}_0)), \sigma_y^2 \mathbf{I})$, we approximate the score of the likelihood as

$$\nabla_{\boldsymbol{z}_t} \log q_t(\boldsymbol{y}|\boldsymbol{z}_t) \approx -\frac{1}{2\sigma_y^2} \nabla_{\boldsymbol{z}_t} \|\mathcal{A}(\mathcal{D}_0(\hat{z}_0(\boldsymbol{z}_t))) - \boldsymbol{y}\|^2. \tag{5}$$

LDPS is a general approach to impose data consistency in the latent domain that works with noisy and potentially nonlinear inverse problems.

## 4 EXPERIMENTS

**Dataset –** We perform experiments on CelebA-HQ ($256 \times 256$) Karras et al. (2018) where we match the training and validation splits used to train LDMs in Rombach et al. (2022), and set aside $1k$ images from the validation split for testing. For comparisons involving image domain score models we test on FFHQ Karras et al. (2019), as pre-trained image-domain score models have been trained on the complete CelebA-HQ dataset, unlike LDMs.

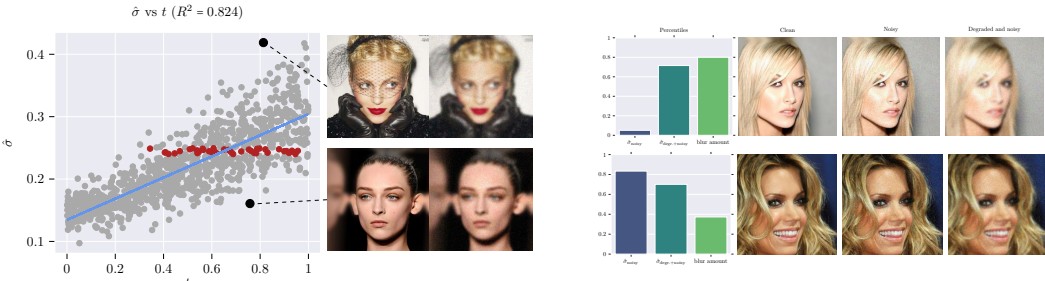

Figure 4: Left: Blur amount ($t$) vs. predicted degradation severity ($\hat{\sigma}$). Outliers indicate that the predicted degradation severity is not solely determined by the amount of blur. The bottom image is 'surprisingly easy' to reconstruct, as it is overwhelmingly smooth with features close to those seen in the training set. The top image is 'surprisingly hard', due to more high-frequency details and unusual features not seen during training. Right: Contributions to predicted severity. Degraded images with approx. the same $\hat{\sigma}$ (red dots on left plot) may have different factors contributing to the predicted severity. The main contributor to $\hat{\sigma}$ in the top image is the image degradation (blur), whereas the bottom image is inherently difficult to reconstruct.

**Degradations –** We investigate three degradations of diverging characteristics. Varying Gaussian blur: we apply Gaussian blur with kernel size of $61$ and sample kernel standard deviation uniformly on $[0, 3]$, where $0$ corresponds to no blurring. Nonlinear blur: we deploy GOPRO motion blur simulated by a neural network model from Tran et al. (2021). This is a nonlinear forward model due to the camera response function. We randomly sample nonlinear blur kernels for each image. Varying random inpainting: we randomly mask $70 - 80\%$ of the pixels, where the masking ratio is drawn uniformly. In all experiments, we add Gaussian noise to images in the $[0, 1]$ range with noise standard deviation of $0.05$.

**Comparison methods –** We compare our method, Flash-Diffusion, with SwinIR Liang et al. (2021), a state-of-the-art Transformer-based supervised image restoration model, DPS Chung et al. (2022a), a diffusion-based solver for noisy inverse problems, and Come-Closer-Diffuse-Faster (CCDF) Chung et al. (2022c), an accelerated image-domain diffusion sampler with two variants: (1) CCDF-DPS: we replace the projection-based data consistency method with diffusion posterior sampling Chung et al. (2022a) to facilitate nonlinear forward models and (2) CCDF-L: we deploy CCDF in latent space using the same LDM as for our method and we replace the data consistency step with LDPS based on (5). The latter method can be viewed as a fixed diffusion steps version of Flash-Diffusion. For all CCDF-variants we use the SwinIR reconstruction as initialization. Finally, we show results of decoding our severity encoder's latent estimate $\hat{z}$ directly without diffusion, denoted by AE (autoencoded). Further details on comparison methods are in Appendix B.

**Models –** We use pre-trained score models from Dhariwal & Nichol (2021) for image-domain diffusion methods[1] and from Rombach et al. (2022) for latent diffusion models[2]. We fine-tune severity encoders from pre-trained LDM encoders and utilize a single convolution layer on top of $\hat{z}$ to predict $\hat{\sigma}$. For more details on the experimental setup and hyperparameters, see Appendix A.

### 4.1 SEVERITY ENCODING

In this section, we investigate properties of the predicted degradation severity $\hat{\sigma}$. We perform experiments on a $1k$-image subset of the validation split. First, we isolate the effect of degradation on $\hat{\sigma}$ (Fig. 3). We fix the clean image and apply increasing amount of Gaussian blur. We observe that $\hat{\sigma}$ is an increasing function of the blur amount applied to the image: heavier degradations on a given image result in higher predicted degradation severity. This implies that the severity encoder learns to capture the amount of information loss caused by the degradation.

Next, we investigate the relation between $\hat{\sigma}$ and the underlying degradation severity (Fig. 4, left). We parameterize the corruption level by $t$, where $t = 0$ corresponds to no blur and additive Gaussian noise ($\sigma = 0.05$) and $t = 1$ corresponds to the highest blur level with the same additive noise. We vary the blur kernel width linearly for $t \in (0, 1)$. We observe that the predicted $\hat{\sigma}$ severities strongly

---

[1] https://github.com/ermongroup/SDEdit
[2] https://github.com/CompVis/latent-diffusion

| Method | Gaussian Deblurring | | | | Nonlinear Deblurring | | | | Random Inpainting | | | |
|---|---|---|---|---|---|---|---|---|---|---|---|---|
| | PSNR(↑) | SSIM(↑) | LPIPS(↓) | FID(↓) | PSNR(↑) | SSIM(↑) | LPIPS(↓) | FID(↓) | PSNR(↑) | SSIM(↑) | LPIPS(↓) | FID(↓) |
| Flash-Diffusion (ours) | 29.16 | 0.8191 | **0.2241** | **29.46** | 27.22 | 0.7705 | **0.2694** | 36.92 | 29.14 | 0.8428 | **0.1948** | 30.36 |
| AE | 29.43 | 0.8366 | 0.2668 | 58.47 | 27.15 | 0.7824 | 0.3351 | 73.81 | 29.18 | 0.8445 | 0.2516 | 56.67 |
| SwinIR Liang et al. (2021) | **30.71** | **0.8598** | 0.2399 | 59.07 | **27.66** | **0.7964** | 0.3072 | 62.11 | **29.96** | **0.8659** | 0.2248 | 51.89 |
| DPS | 28.35 | 0.7806 | 0.2470 | 55.17 | 22.82 | 0.6247 | 0.3603 | 72.20 | 28.22 | 0.8057 | 0.2462 | 56.01 |
| CCDF-DPS Chung et al. (2022c) | 30.02 | 0.8365 | 0.2324 | 50.62 | 26.98 | 0.7445 | 0.2840 | 56.92 | 29.02 | 0.8237 | 0.2344 | 51.86 |
| CCDF-L | 29.55 | 0.8377 | 0.2346 | 49.06 | 27.25 | 0.7793 | 0.2833 | 55.85 | 28.98 | 0.8496 | 0.2092 | 43.35 |

Table 1: Experimental results on the FFHQ test split.

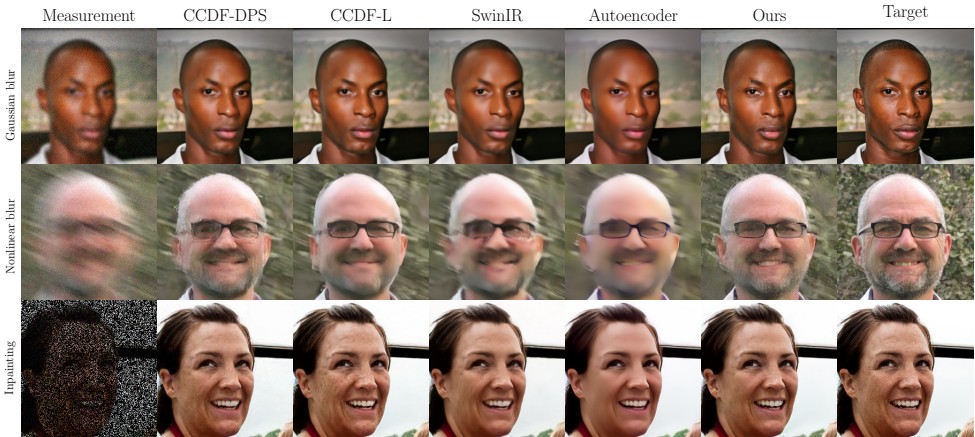

Figure 5: Visual comparison of FFHQ reconstructions under varying levels of Gaussian blur (top row), nonlinear motion blur (middle row), and varying amounts of random inpainting (bottom row). We add Gaussian noise ($\sigma = 0.05$) to the measurements. See more examples in Appendix H.

correlate with the corruption level. However, the existence of outliers suggest that factors other than the corruption level may also contribute to the predicted severities. The bottom image is predicted to be 'surprisingly easy', as other images of the same corruption level are typically assigned higher predicted severities. This sample is overwhelmingly smooth, with a lack of fine details and textures, such as hair, present in other images. Moreover, the image shares common features with others in the training set. On the other hand, the top image is considered 'surprisingly difficult', as it contains unexpected features and high-frequency details that are uncommon in the dataset. This example highlights the potential application of our technique to hard example mining and dataset distillation.

Finally, we analyze the contribution of different factors to the predicted degradation severities (Fig. 4, right). We apply severity encoding to both the clean image with noise (no blur) and the noisy and degraded image, resulting in predicted severities $\hat{\sigma}_{noisy}$ and $\hat{\sigma}_{degr.+noisy}$. We quantify the difficulty of samples relative to each other via percentiles of the above two quantities, where we use $\hat{\sigma}_{noisy}$ as a proxy for the difficulty originating from the image structure. We observe that for a fixed $\hat{\sigma}_{degr.+noisy}$, the composition of degradation severity may greatly vary. The two presented images have been assigned approximately the same $\hat{\sigma}_{degr.+noisy}$, however the top image is inherently easy to encode (low $\hat{\sigma}_{noisy}$ percentile) compared to other images in the dataset, therefore the severity is mostly attributed to the image degradation. On the other hand, the bottom image with the same $\hat{\sigma}_{degr.+noisy}$ is less corrupted by blur, but with high $\hat{\sigma}_{noisy}$ indicating a difficult image. This example further corroborates the interaction between ground truth signal structure and the applied corruption in determining the difficulty of a reconstruction task.

## 4.2 SAMPLE-ADAPTIVE RECONSTRUCTION

**Comparison with state-of-the-art –** Table 1 summarizes our experimental results comparing various techniques. We observe that Flash-Diffusion consistently outperforms other diffusion-based solvers in terms of perceptual metrics such as LPIPS and FID. SwinIR, a state-of-the-art supervised image restoration model achieves higher PSNR and SSIM as diffusion methods, however the reconstructions lack detail compared to other techniques. This phenomenon is due to the perception-distortion trade-off Blau & Michaeli (2018): improving perceptual image quality is fundamentally at odds with distortion metrics. Furthermore, we highlight that our initial reconstructions are worse than SwinIR reconstructions (compare AE vs. SwinIR) used to initialize CCDF-variants. Despite this, we still achieve overall better perceptual reconstruction quality.

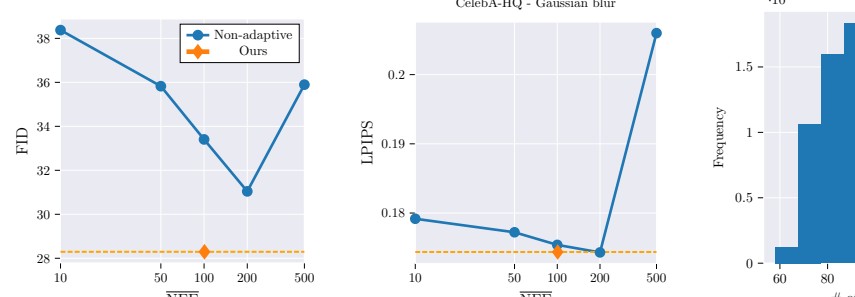

Figure 6: Comparison of sample-adaptive reconstruction (ours) with CCDF-L (non-adaptive, i.e. fixed starting time $N'$ for all samples). Left and center: We plot the average number of diffusion steps ($\overline{\text{NFE}}$) performed by our algorithm vs. CCDF-L with various starting times ($N' \in [10, 50, 100, 200, 500]$). Flash-Diffusion achieves the best FID and near-optimal LPIPS compared to CCDF-L with any choice of non-adaptive starting time. Right: We plot the histogram of predicted starting times for our algorithm. The spread around the mean highlights the adaptivity of Flash-Diffusion.

We perform visual comparison of reconstructed samples in Figure 5. We observe that Flash-Diffusion can reconstruct fine details, significantly improving upon the autoencoded reconstruction used to initialize the reverse diffusion process. SwinIR produces reliable reconstructions, but with less details compared to diffusion-based methods. Moreover, note that diffusion-based solvers with fixed number of diffusion steps tend to under-diffuse (see 2nd row, lack of details) or over-diffuse (4th row, high-frequency artifacts) leading to subpar reconstructions.

**Efficiency of the method** – In order to demonstrate the gains in terms of reconstruction efficiency, we compare Flash-Diffusion to CCDF-L across various (fixed) number of reverse diffusion steps (Fig. 6). We observe that our adaptive method achieves the best FID across any fixed number of steps by a large margin. Moreover, it achieves near-optimal LPIPS with often $2\times$ less average number of diffusion steps. We observe that the predicted diffusion steps are spread around the mean and not closely concentrated, further highlighting the adaptivity of our proposed method.

**Robustness and limitations–** The performance of Flash-Diffusion relies on the accuracy of the severity encoder trained in a supervised fashion on the degraded image distribution. Therefore, it is crucial to understand the robustness of severity encoding to noise level and forward model shifts in test time. In terms of noise robustness, we observe that severity encoding maintains its performance on noise level lower than in the training setting. At higher noise levels, severity encoder estimates deteriorate, however the performance degradation of Flash-Diffusion is more graceful than in case of other supervised techniques. More details can be found in Appendix D.1. In terms of forward model mismatch, we find that as long as the train and test forward operators are similar (heavy Gaussian blur and nonlinear blur), severity encoding can maintain acceptable performance. However in case of significant mismatch, the accuracy of severity encoding breaks down. An in-depth characterization of this phenomenon is detailed in Appendix D.2.

## 5 CONCLUSIONS

In this work, we make the key observation that the difficulty of solving an inverse problem may vary greatly on a sample-by-sample basis, depending on the ground truth signal structure, the applied corruption, the model, the training set and the complex interactions between these factors. Despite this natural variation in the difficulty of a reconstruction task, most techniques apply a rigid model that expends the same compute irrespective of the amount of information destroyed in the noisy measurements, resulting in suboptimal performance and wasteful resource allocation. We propose Flash-Diffusion, a sample-adaptive method that predicts the degradation severity of corrupted signals, and utilizes this estimate to automatically tune the compute allocated to reconstruct the sample. In particular, we use the prediction error of an encoder in latent space as a proxy for reconstruction difficulty, which we call severity encoding. Then, we leverage a latent diffusion process to reconstruct the sample, where the length of the sampling trajectory is directly scaled by the predicted severity. We experimentally demonstrate that the proposed technique achieves performance on par with state-of-the-art diffusion-based reconstruction methods, but with greatly improved compute efficiency.

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

# A  TRAINING DETAILS

Here we provide additional details on the training setup and hyperparameters.

**Model architecture –** In all experiments, we use an LDM model pre-trained on the CelebA-HQ dataset out of the box. We fine-tune the severity encoder from the LDM's pre-trained encoder. We obtain the degradation severity estimate $\hat{\sigma} \in \mathbb{R}^+$ from the latent reconstruction $\hat{z} \in \mathbb{R}^d$ as

$$\hat{\sigma} = \frac{1}{d}\sum_{i=1}^{d}[Conv(\hat{z})^2]_i,$$

where $Conv(\cdot)$ is a learned $1 \times 1$ convolution with $d$ input and $d$ output channels.

**Training setup –** We train severity encoders using Adam optimizer with batch size 28 and learning rate $0.0001$ for about $200k$ steps until the loss on the validation set converges. We use Quadro RTX 5000 and Titan GPUs.

**Hyperparameters –** We scale the reconstruction loss terms with their corresponding dimension ($d$ for $L_{lat.rec.}$ and $n$ for $L_{im.rec.}$), which we find to be sufficient without tuning for $\lambda_{im.rec.}$. We tune $\lambda_\sigma$ via grid search on $[0.1, 1, 10]$ on the varying Gaussian blur task and set to 10 for all experiments.

For latent diffusion posterior sampling, following Chung et al. (2022a) we scale the posterior score estimate with the data consistency error as

$$\nabla_{\boldsymbol{z}_t} \log q_t(\boldsymbol{z}_t|\boldsymbol{y}) \approx s_\theta(\boldsymbol{z}_t) - \eta_t \nabla_{\boldsymbol{z}_t} \|\mathcal{A}(\mathcal{D}_0(\hat{\boldsymbol{z}}_0(\boldsymbol{z}_t))) - \boldsymbol{y}\|^2,$$

where

$$\eta_t = \frac{\eta}{\|\mathcal{A}(\mathcal{D}_0(\hat{\boldsymbol{z}}_0(\boldsymbol{z}_t))) - \boldsymbol{y}\|}, \tag{6}$$

and $\eta > 0$ is a tuning parameter. We perform grid search over $[0.5, 1.0, 1.5, 2.0, 5.0, 7.0, 10.0]$ over a small subset of the validation set (100 images) and find $\eta = 1.0$ to work the best for Gaussian and nonlinear blur and $\eta = 7.0$ for random inpainting.

Similarly, we tune the noise correction parameter $c$ on the same validation subset by grid search over $[0.5, 0.8, 1.0, 1.2, 1.5]$ and find $c = 1.2$ for Gaussian blur and random inpainting and $c = 1.0$ for nonlinear blur to work the best.

# B  COMPARISON METHOD DETAILS

For all methods, we use the train and validation splits provided for CelebA and FFHQ in the GitHub repo of "Taming Transformers" [3]. For the test split, we subsample 1000 ids from the corresponding validation ids file. Specific ids we used will be available when the codebase is released. We provide the specifics for each comparison method next.

**SwinIR:** For all experiments, we train SwinIR using Adam optimizer with batch size 28 for 100 epochs. We use learning rate $0.0002$ for the first 90 epochs and drop it by a factor of 10 for the remaining 10 epochs. We use Quadro RTX 5000 and Titan GPUs.

**CCDF-DPS:** We implement this method by modifying the official GitHub repo [4] of DPS Chung et al. (2022a). Instead of projection based conditioning, we replace it with the DPS updates to handle noise in the measurements and nonlinear degradations. As the initial estimate, we use the output of SwinIR model that we trained. We tune the data consistency step size $\zeta'$ and number of reverse diffusion steps $N'$ by doing 2D grid search over $[0.5, 1.0, 2.0, 3.0, 5.0, 10.0, 20.0] \times [1, 10, 20, 50, 100, 200, 1000]$ on the small subset of validation split of FFHQ dataset (100 images) based on LPIPS metric. For Gaussian blur, we find the optimal hyperparameters to be $(\zeta', N') = (5.0, 20)$, for nonlinear blur $(\zeta', N') = (3.0, 100)$ and for random inpainting $(\zeta', N') = (3.0, 50)$.

**CCDF-L:** This method is the same as ours but with a fixed starting time and initial estimate provided by SwinIR model we trained. We tune the data consistency step size $\eta$ and the number of reverse diffusion steps $N'$ by doing grid search over $[0.5, 1.0, 1.5, 2.0, 5.0, 7.0, 10.0] \times [20, 50, 100, 200]$ on the small subset of validation split of FFHQ (100 images) based on LPIPS metric. For varying blur

---

[3]https://github.com/CompVis/taming-transformers/tree/master/data
[4]https://github.com/DPS2022/diffusion-posterior-sampling

experiments, we found the optimal value to be $(\eta, N') = (1.0, 100)$, for nonlinear blur $(\eta, N') = (1.5, 200)$ and for random inpainting $(\eta, N') = (10.0, 100)$.

**DPS:** This method can be seen as a special case of CCDF-DPS where number of reverse diffusion steps is fixed to $N' = 1000$. From the same 2D grid search we performed for CCDF-DPS, we find the optimal data consistency step size $\zeta'$ to be $5.0$ for Gaussian blur, $0.5$ for nonlinear blur and $3.0$ for random inpainting.

**Autoencoded (AE):** We use the latent at severity encoders output and decode it without reverse diffusion to get the reconstruction.

## C  ABLATION OF FLASH-DIFFUSION COMPONENTS

Flash-Diffusion combines sample-by-sample adaptation of the sampling trajectory via severity encoding with latent-space diffusion. We ablate the effect of these two key components and demonstrate their relative contribution to the performance of Flash-Diffusion. In particular, we ablate the effect of adaptation by running CCDF-L with the same (fixed) number of reverse diffusion steps as the average of Flash-Diffusion sampling steps on the corresponding task ($N = 100$ for Gaussian deblurring, $N = 134$ for nonlinear deblurring). We further ablate the effect of latent domain reconstruction via repeating the above experiment with a pixel-space score model (CCDF-DPS). We optimized all models for best performance in terms of LPIPS via grid search over $\eta = [0.5, 1.0, 1.5, 2.0, 3.0, 5.0, 20.0]$ and selected $\eta = 1.5$ for CCDF-L and $\eta = 3.0$ for CCDF-DPS. Our findings are summarized in Table 2. We observe, that adaptation via severity encoding provides a significant boost to the performance in terms of our target metric, LPIPS. We obtain similar results with and without latent space diffusion, however latent diffusion is preferable since (1) it has greatly reduced computational cost and (2) degradation severity is better captured in latent space.

| | | Gaussian Deblurring | | | | Nonlinear Deblurring | | | |
|---|---|---|---|---|---|---|---|---|---|
| Adaptive? | Latent? | PSNR($\uparrow$) | SSIM($\uparrow$) | LPIPS($\downarrow$) | FID($\downarrow$) | PSNR($\uparrow$) | SSIM($\uparrow$) | LPIPS($\downarrow$) | FID($\downarrow$) |
| ✓ | ✓ | 29.16 | 0.8191 | **0.2241** | **29.46** | 27.22 | 0.7705 | **0.2694** | **36.92** |
| ✗ | ✓ | **29.55** | **0.8377** | 0.2346 | 49.06 | **27.35** | **0.7844** | 0.2847 | 54.25 |
| ✗ | ✗ | 29.23 | 0.8058 | 0.2377 | 54.20 | 26.83 | 0.7379 | 0.2843 | 57.82 |

Table 2: Ablation of different components of the Flash-Diffusion framework on the FFHQ dataset.

## D  ROBUSTNESS STUDIES

### D.1  NOISE ROBUSTNESS

**Robustness of severity encoding –** As demonstrated throughout the paper, our severity encoder provides useful estimates of the degradation severity of samples in the scenario where the degradation and noise level matches that in the training setup. However, it is very important to understand the limitations of severity encoding in the presence of test-time shifts in the corruption process.

Our most crucial expectation towards the severity encoder is to predict a degradation severity $\hat{\sigma}(\boldsymbol{y})$ that is a non-decreasing function of the true underlying corruption level. More specifically, assume that the forward model $\mathcal{A}$ is parameterized by a severity level $s \in [0, 1]$, with $s = 0$ being least severe, $s = 1$ being most severe. For instance, the standard deviation of the Gaussian blur kernel may serve as $s$ in case of Gaussian blur (see a more rigorous treatment of degradation severity in Fabian et al. (2023)). Then for a given specific clean image $\boldsymbol{x}_0$ and its corrupted versions $\boldsymbol{y}_{s'} := \mathcal{A}(\boldsymbol{x}_0; s') + \boldsymbol{n}, \ \boldsymbol{y}_{s''} := \mathcal{A}(\boldsymbol{x}_0; s'') + \boldsymbol{n}$, we expect our severity encoder to satisfy

$$\hat{\sigma}(\boldsymbol{y}_{s'}) \leq \hat{\sigma}(\boldsymbol{y}_{s''}), \quad , \quad \forall s', s'' \in [0, 1], \ s' < s'', \tag{7}$$

which we call the *monotonicity requirement*, i.e. the ordering of predicted severities need to match the ordering of the true underlying severity levels.

We demonstrate this property in Figure 3 for the case where the measurement noise level corresponds to the training setting ($\sigma_{\boldsymbol{y}} = 0.05$). Here, we observe that the predicted severity is a non-decreasing

function of the true blur level (monotonicity requirement is satisfied) when the ground truth clean image is fixed. Next, we increase the measurement noise in test time and investigate how the monotonicity requirement breaks down (see Figure 7). At $\sigma_y = 0.1$, which correspond to an increase in noise variance by a factor of $4\times$ compared to the training setting, the severity encoder fails to satisfy the monotonicity requirement (Fig. 7b), but still manages to provide useful estimates in some regimes (high blur). This is due to the fact that (1) the severity encoder has not encountered such high noise levels during training and thus the estimates are highly inaccurate, and (2) the high noise 'over-powers' the effect of blurring and therefore it becomes more challenging to distinguish between images corrupted by similar blur amounts. Finally, as we increase the test-time measurement noise to $\sigma_y = 0.2$ (variance $16\times$ training setup), the severity encoder completely breaks down, and produces a similar severity estimate for all blur amounts (Fig. 7c).

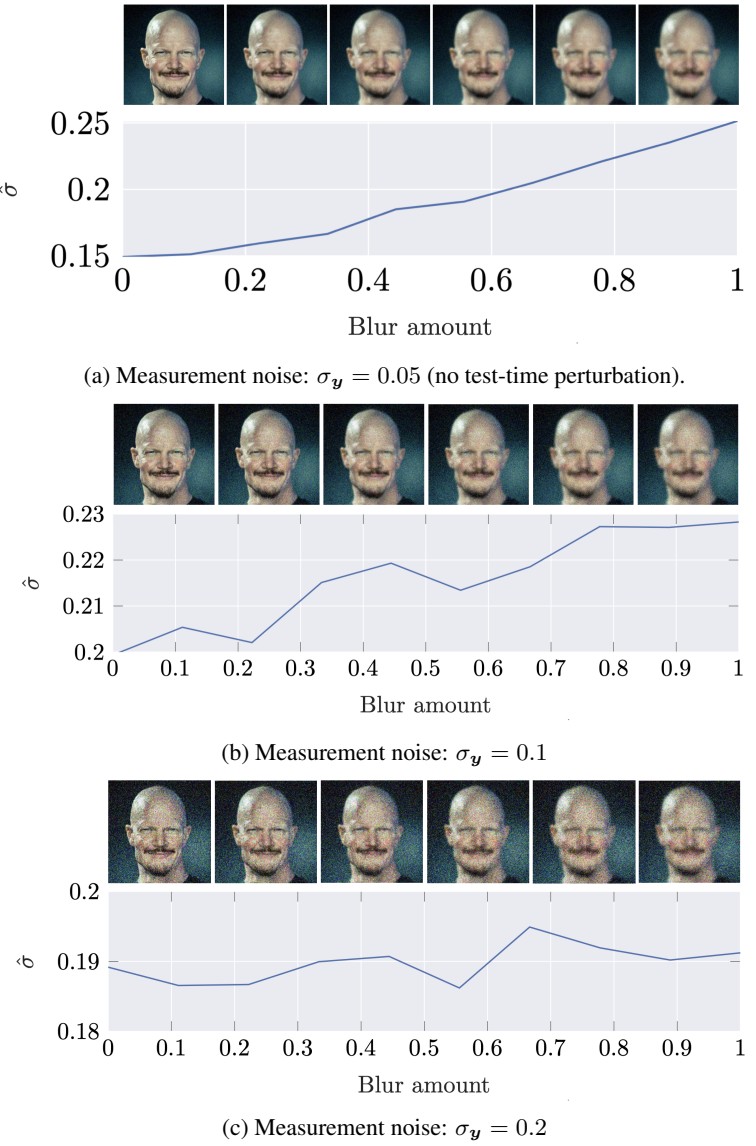

(a) Measurement noise: $\sigma_y = 0.05$ (no test-time perturbation).

(b) Measurement noise: $\sigma_y = 0.1$

(c) Measurement noise: $\sigma_y = 0.2$

Figure 7: Effect of perturbation in the measurement noise level in test time on the predictions of the severity encoder. We expect the predicted severity to be a non-decreasing function of the true blur amount, as depicted in Figure 7a. At high noise level perturbations the severity encoder violates the monotonicity requirement (Figure 7b) and eventually breaks down completely at extreme measurement noise variances (Figure 7c).

To get an even more detailed understanding of the limitations of the severity encoder, we introduce the notion of *ordering accuracy* that qualitatively captures how well the monotonicity requirement is satisfied.

**Definition 1** (Ordering accuracy). *Let $\boldsymbol{x}_0$ be a clean image, and $\boldsymbol{y}_{s_1}, \boldsymbol{y}_{s_2}, ..., \boldsymbol{y}_{s_n}$ a sequence of corrupted observations of $\boldsymbol{x}_0$, such that $\boldsymbol{y}_{s_i} = \mathcal{A}(\boldsymbol{x}_0; s_i) + \boldsymbol{n}$, and $0 \leq s_1 < s_2 < ... < s_n \leq 1$, where $s_i$ parameterizes the severity level of the degradation (higher the more severe). Let $\hat{\sigma}_i := \hat{\sigma}(\boldsymbol{y}_{s_i})$ denote the corresponding estimates of degradation severity from the severity encoder, and*

$$g(\boldsymbol{y}_{s_i}, \boldsymbol{y}_{s_j}; \boldsymbol{x}_0) = \begin{cases} 1 & if \ \ sign(s_i - s_j) = sign(\hat{\sigma}_i - \hat{\sigma}_j) \\ 0 & otherwise \end{cases} \tag{8}$$

*Then, the ordering accuracy (OA) of the severity encoder on sample $\boldsymbol{x}_0$ and severity levels $s_1, ..., s_n$ under the degradation model $\mathcal{A}(\cdot, s)$ is*

$$OA(\boldsymbol{x}_0; \boldsymbol{y}_{s_1}, ..., \boldsymbol{y}_{s_n}) := \frac{1}{n(n-1)} \sum_{i \neq j} g(\boldsymbol{y}_{s_i}, \boldsymbol{y}_{s_j}; \boldsymbol{x}_0) \tag{9}$$

Intuitively, the ordering accuracy takes a sequence of increasingly degraded images and the corresponding severity estimates from the encoder and evaluates the portion of pairs of degraded images for which the ordering of ground truth degradation level matches the ordering of predicted severities. Ideally, if the monotonicity requirement is satisfied, we have ordering accuracy of 1 for every sample. On the other hand, if the severity estimates are generated randomly we have ordering accuracy of $0.5$ (the probability that any pair has the correct ordering is $0.5$).

We leverage the notion of ordering accuracy to evaluate the degradation of severity encoding performance under test-time measurement noise perturbations. In particular, we consider the Gaussian deblurring task and select $n = 10$ blur kernel stds over a uniform grid in $[0.0, 3.0]$, with $s_1 = 0$ corresponding to no blurring and $s_n = 1$ corresponding to blurring with blur kernel std of $3.0$. In Figure 8, we plot how the average ordering accuracy evolves as we change the measurement noise level evaluated on 100 validation samples on the CelebA dataset. We observe that for noise levels up to the training noise level, the severity encoder has perfect ordering accuracy. That is, we can expect reliable performance on noise levels that are different but lower than the training measurement noise. This property indicates that it may be beneficial to train the severity encoder at higher noise levels than the expected test-time noise in order to increase robustness. As the noise level perturbation increases, severity encoding performance drops, approaching random guessing (ordering accuracy of $0.5$) at extreme noise variances.

**Robustness of Flash-Diffusion performance –** Next, we investigate the impact of noise level mismatch in test-time on the reconstruction performance of Flash-Diffusion. In particular, we perform experiments on our best Flash-Diffusion model trained on the CelebA dataset for the Gaussian deblurring task with fixed measurement noise of $\sigma_{\boldsymbol{y}} = 0.05$. We vary the measurement noise in test time and compare the reconstruction performance of Flash-Diffusion to CCDF-L and SwinIR. We depict the results of the robustness study in Figure 9. We observe that both in terms of perceptual and distortion metrics, the performance of Flash-Diffusion degrades more gracefully compared to CCDF-L and SwinIR. We hypothesize that this additional robustness is due to the sample-by-sample adaptivity of Flash-Diffusion. As the noise level increases, our method can automatically adapt by increasing the number of diffusion steps due to higher degradation severity predictions from the severity encoder. However, other techniques expend the same compute even though the corruption level is significantly increased in test time.

The adaptation to higher measurement noise levels in test time is demonstrated in Figure 10. Even though the severity encoder has not been trained on noise levels other than $\sigma_{\boldsymbol{y}} = 0.05$, its predictions are still useful enough to adapt the sampling trajectory to the changes in noise level, up to a limit. At noise levels significantly higher than in the training setting ($\sigma_{\boldsymbol{y}} = 0.1$ and above), the performance of the severity encoder breaks down and is unable to provide accurate severity estimations. This phenomenon is corroborated by our observation of degrading ordering accuracy at high noise levels (see Figure 8 and Figure 7b).

## D.2 ROBUSTNESS AGAINST FORWARD MODEL MISMATCH

Our method relies on a severity encoder that has been trained on paired data of clean and degraded images under a specific forward model. We simulate a mismatch between the severity encoder

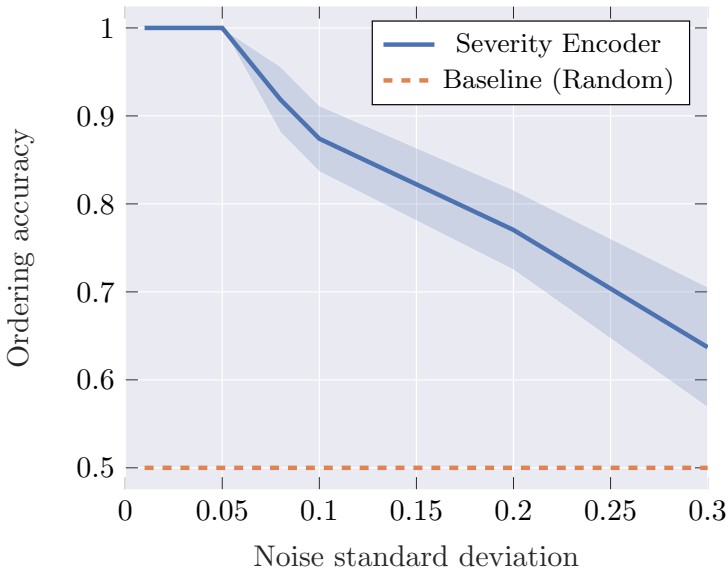

Figure 8: Evaluation of severity encoding performance under test-time noise level perturbation. The training noise level is $\sigma_y = 0.05$. The severity encoder is robust to arbitrary reductions in noise level in test time (no drop in ordering accuracy). As the noise level perturbation increases, performance degrades and approaches random guessing (ordering accuracy of 0.5). Mean over 100 validation samples and 3 measurement generating random seeds is plotted, with $\pm$ standard deviation over random seeds as shaded area.

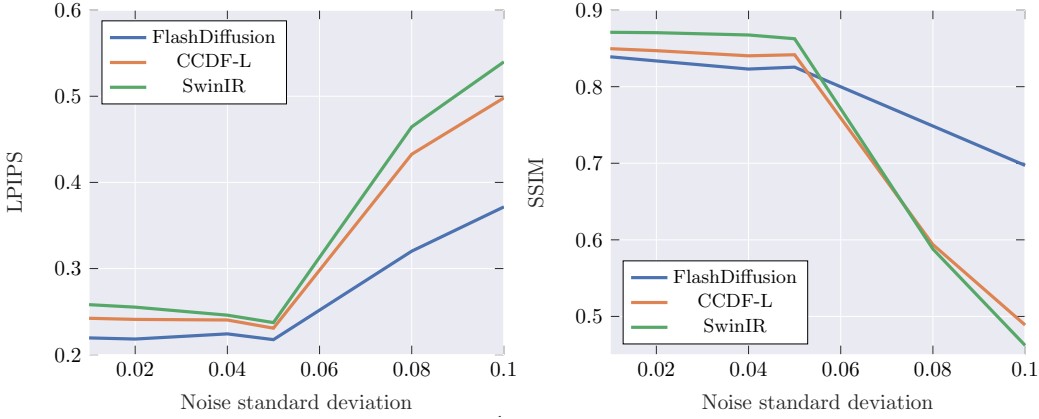

Figure 9: Performance of Flash-Diffusion and other methods under measurement noise level perturbation in test time. The performance of Flash-Diffusion degrades more gracefully compared with competing methods due to sample-by-sample adaptivity.

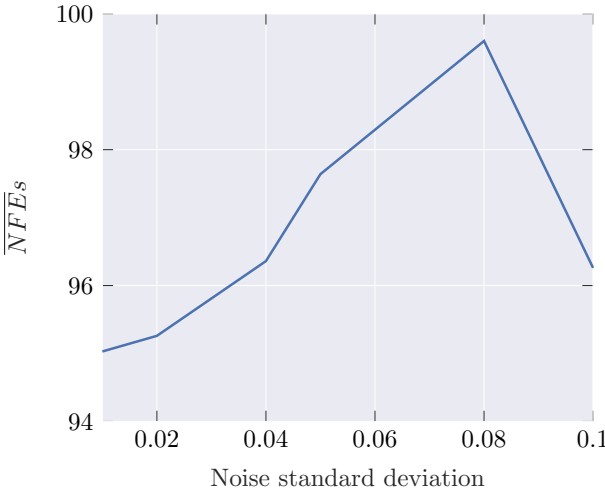

Figure 10: Due to the adaptivity of Flash-Diffusion, the length of the sampling trajectory is automatically scaled to the input noise level in test time, even in case of noise levels different from the training setup. At extreme noise discrepancies ($\sigma_y = 0.1$) the severity encoder breaks down and unable to adapt to the increased degradation severity.

fine-tuning operator and test-time operator in order to investigate the robustness of our technique with respect to forward model perturbations. In particular, we run the following experiments to assess the test-time shift: 1) we train the encoder on Gaussian blur and test on nonlinear blur and 2) we train the encoder on nonlinear blur and test on Gaussian blur.

**Robustness of severity encoding** – First, following our methodology in Appendix D.1, we investigate how severity encoding performance degrades if the image degradation forward model is different from what the severity encoder has been trained on. We train the severity encoder on nonlinear blur and evaluate it on Gaussian blur of varying amounts on the CelebA dataset. We expect the predicted severity to be a non-decreasing function of the true blur amount. However, as Figure 11 demonstrates, the monotonicity requirement is violated due to the forward model mismatch. In particular, the severity encoder fails to predict the degradation severities of lightly blurred images. However, we observe satisfactory performance at high blur amounts (above 0.3 in Fig. 11). We hypothesize that at high blur levels, Gaussian blur and nonlinear blur become similar, thus the severity encoder trained on either may provide acceptable predictions for both. We further support this hypothesis by evaluating the ordering accuracy on a validation set of 100 CelebA images. The results are summarized in Table 3. We observe that the ordering accuracy is poor (mean OA = 0.7704) when evaluated across the full range of blur levels. However, it is near-perfect when low blur levels are excluded from the evaluation (mean OA = 0.9778). This experiment highlights the severity encoder's potential for generalization to unseen forward models, which opens up interesting future directions for fast adaptation to new tasks.

| Blur range | Avg. OA |
|---|---|
| [0, 1] | 0.7704 |
| [0.3, 1] | 0.9778 |

Table 3: Ordering accuracy of a severity encoder trained on nonlinear blur and evaluated on Gaussian blur of varying blur amount. The severity encoder provides fair estimates even under forward model mismatch in the high blur regime.

**Robustness of Flash-Diffusion performance** – The results on the FFHQ test set are in Table 4. We observe minimal loss in performance when nonlinear blur encoder is used for reconstructing images corrupted by Gaussian blur. For the nonlinear deblurring task, using Gaussian blur encoder results in a more significant drop in the performance, while still providing acceptable reconstructions. These results are expected, as Gaussian blur can be thought of as a special case of the nonlinear blur model we consider. Therefore even when the encoder is swapped, it can provide meaningful mean and

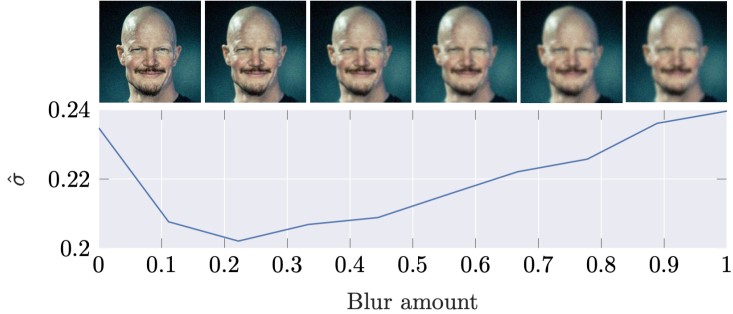

Figure 11: Effect of perturbation in the forward model in test time on the predictions of the severity encoder. We train the severity encoder on nonlinear blur and evaluate it on Gaussian blur of varying amounts. We expect the predicted severity to be a non-decreasing function of the true blur amount. However, due to the forward model mismatch the monotonicity requirement is violated for low blur levels.

error estimation. However, the Gaussian blur encoder has never been trained on images corrupted by nonlinear blur. As such, the mean estimate is worse, resulting in a larger performance drop. Note that we did not re-tune the hyper-parameters in these experiments and doing so may potentially alleviate the loss in performance.

| Method | Gaussian Deblurring | | | | Nonlinear Deblurring | | | |
|---|---|---|---|---|---|---|---|---|
| | PSNR($\uparrow$) | SSIM($\uparrow$) | LPIPS($\downarrow$) | FID($\downarrow$) | PSNR($\uparrow$) | SSIM($\uparrow$) | LPIPS($\downarrow$) | FID($\downarrow$) |
| Ours + Gaussian blur encoder | 29.16 | 0.8191 | 0.2241 | 29.467 | 25.36 | 0.7238 | 0.3416 | 54.90 |
| Ours + NL blur encoder | 28.96 | 0.8129 | 0.2362 | 30.34 | 27.22 | 0.7705 | 0.2694 | 36.92 |

Table 4: Robustness experiments on the FFHQ test split.

# E DDIM EXPERIMENTS

Throughout the paper, we demonstrated the performance of Flash-Diffusion built upon DDPM Ho et al. (2020) sampling with N=1000, however accelerated sampling schemes such as DDIM Song et al. (2021a) have demonstrated comparable generation performance to DDPM with significantly less reverse diffusion steps. Our proposed framework can be paired with any diffusion-based sampling technique as long as the SNR can be evaluated analytically for any time step (see (4)). In this section we investigate the performance of Flash-Diffusion, when paired with DDIM sampling. The DDIM updates in latent domain take the form

$$z_{i-1} = \sqrt{\bar{\alpha}_{i-1}}\hat{z}_0(z_t) + \sqrt{1 - \bar{\alpha}_{i-1} - \delta^2\tilde{\beta}_i^2}\epsilon_\theta(z_i, i) + \delta\tilde{\beta}_i^2\epsilon,$$

where $\hat{z}_0(z_t)$ is the estimator of the posterior mean based on Tweedie's formula, $\tilde{\beta}_i = \sqrt{(1 - \bar{\alpha}_{i-1})/(1 - \bar{\alpha}_i)}\sqrt{1 - \bar{\alpha}_i/\bar{\alpha}_{i-1}}$ and $\delta$ parameter controlling the stochasticity of the sampling process. For $\delta = 0.0$ one obtains a fully deterministic inference procedure, while $\delta = 1.0$ corresponds to the ancestral sampling of DDPM.

We run experiments on the FFHQ test set under Gaussian blur degradation of varying magnitude matching the setup in Section 4. We investigate DDIM sampling with N=20 and N=100 steps and for both $\delta = 0.0$ (deterministic) and $\delta = 1.0$ (DDPM-like). We tune the data consistency step size $\eta$ in (6) by grid search over $[0.1, 0.5, 1.0, 1.5]$ on a validation split of 100 images of FFHQ for each combination of N and $\delta$. We select the value $\eta = 1.5$ as it provides the best LPIPS across all experiments.

The results are shown in Table 5. First, we observe that combining Flash-Diffusion with DDIM sampling results in vastly accelerated sampling: we obtain reasonable reconstructions in 10.4 (N=100) or 2.5 (N=20) diffusion steps on average over the test set. Image quality in terms of distortion and

perceptual metrics is comparable with other diffusion based solvers under DDIM sampling with N=100 steps, but with compute cost reduced by an order of magnitude compared to our baseline Flash-Diffusion algorithm. However, reconstruction quality degrades significantly when deploying N=20 DDIM sampling. We hypothesize that the LDPS step size scheduling defined in (6) needs to be tailored to DDIM, which is an interesting direction for future work. Finally, we note that we find Flash-Diffusion performance to be robust to the setting of $\delta$ in the DDIM sampler, as we obtain very similar results for $\delta = 0.0$ and $\delta = 1.0$.

Next, we demonstrate the importance of leveraging the sampling trajectory adaptation provided by severity encoding. In fact, one could argue that the reduction in sampling steps of Flash-Diffusion can be achieved using complete sampling trajectories with DDIM updates (without the shortcut initialization from the severity encoder). In particular, we experiment with LDPS with DDIM sampling of N=100. This closely matches the average number of steps Flash-Diffusion uses for sampling (FFHQ + Gaussian deblur task, see top row of Figure 6), but instead of jumping ahead in the reverse process at initialization, it starts from the i.i.d. Gaussian distribution similar to DPS. The results are shown in Table 6. We find that LDPS with DDIM sampling far underperforms Flash-Diffusion and other diffusion-based reconstruction techniques. In fact, we observe a large variation in reconstruction quality, from high fidelity reconstructions to images with heavy artifacts. We hypothesize that this effect is due to large errors in the posterior mean estimate in the early stages of diffusion resulting in inaccurate posterior sampling, which is further compounded by the large variance in sample degradations.

| | | $\delta = 0.0$ | | | | $\delta = 1.0$ | | | |
|---|---|---|---|---|---|---|---|---|---|
| Sampling | Avg. NFE | PSNR($\uparrow$) | SSIM($\uparrow$) | LPIPS($\downarrow$) | FID($\downarrow$) | PSNR($\uparrow$) | SSIM($\uparrow$) | LPIPS($\downarrow$) | FID($\downarrow$) |
| Ours (DDPM, N=1000) | 100.1 | - | - | - | - | 29.16 | 0.8191 | 0.2241 | 29.46 |
| Ours (DDIM, N=100) | 10.4 | 28.04 | 0.7956 | 0.2537 | 32.08 | 27.70 | 0.7901 | 0.2550 | 31.83 |
| Ours (DDIM, N=20) | 2.5 | 26.05 | 0.7482 | 0.3204 | 57.77 | 26.09 | 0.7504 | 0.3168 | 54.88 |

Table 5: Comparison of sampling techniques in the Flash-Diffusion framework on FFHQ Gaussian deblurring.

| | Gaussian Deblurring | | | | Nonlinear Deblurring | | | |
|---|---|---|---|---|---|---|---|---|
| Sampling | PSNR($\uparrow$) | SSIM($\uparrow$) | LPIPS($\downarrow$) | FID($\downarrow$) | PSNR($\uparrow$) | SSIM($\uparrow$) | LPIPS($\downarrow$) | FID($\downarrow$) |
| LDPS, DDIM N=100, $\delta = 0.0$ | 16.62 | 0.4192 | 0.6224 | 233.74 | 13.52 | 0.3694 | 0.6472 | 278.25 |
| LDPS, DDIM N=100, $\delta = 1.0$ | 18.23 | 0.4771 | 0.5474 | 129.67 | 14.96 | 0.4071 | 0.5533 | 86.68 |

Table 6: LDPS without severity encoding using DDIM sampling (N=100) on the FFHQ test split.

## F    ADDITIONAL EFFICIENCY EXPERIMENTS

We perform additional experiments on the nonlinear blurring task to demonstrate the efficiency of Flash-Diffusion compared to non-adaptive baselines. Figure 12 indicates that the adaptive sampling of Flash-Diffusion achieves better perceptual quality than any fixed-length sampling trajectory. The adaptivity is further supported by the wide spread of sampling steps utilized by Flash-Diffusion.

## G    LIMITATIONS

We identify the following limitations of our framework.

1. The proposed reconstruction method requires degraded-clean image pairs for fine-tuning the severity encoder. Fine-tuning has to be performed separately for each degradation, thus the method is less flexible than DPS and similar diffusion solvers. However, we argue that the fine-tuning step has fairly low cost and greatly pays off in reconstruction performance and efficiency.

2. The proposed severity estimation method breaks down at high noise perturbations compared to the training settings and when there is a significant test-time shift in the forward model.

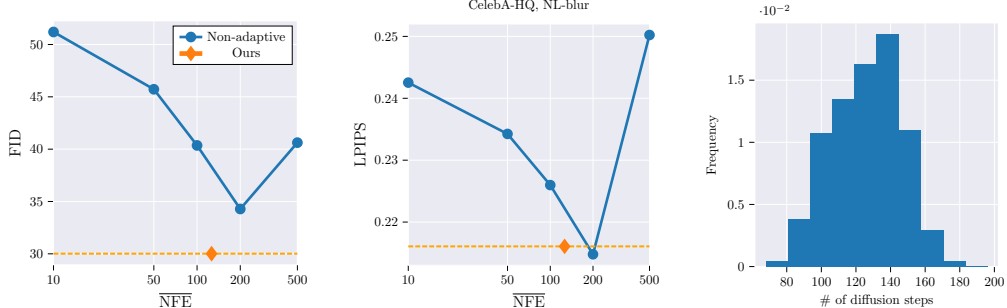

Figure 12: Comparison of sample-adaptive reconstruction (ours) with CCDF-L (non-adaptive, i.e. fixed starting time $N'$ for all samples) on the nonlinear blur task. Left and center: We plot the average number of reverse diffusion steps ($\overline{\text{NFE}}$) performed by our algorithm vs. CCDF-L with various starting times ($N' \in [10, 50, 100, 200, 500]$). Flash-Diffusion achieves the best FID and near-optimal LPIPS compared to CCDF-L with any choice of non-adaptive starting time. Right: We plot the histogram of predicted starting times for our algorithm. The spread around the mean highlights the adaptivity of our proposed technique.

3. The assumption of i.i.d. Gaussian prediction error provides a simple way to estimate the severity, however does not necessarily hold in practice. We believe that more realistic error models can further improve our technique, which we leave for future work.

4. Latent diffusion posterior sampling has some additional compute cost compared to DPS due to differentiating through the decoder. Maintaining data-consistency through only latent space information would alleviate this issue. However, it is not clear how to do this for arbitrary degradations, which we leave for future work.

# H    ADDITIONAL SAMPLES FOR VISUAL COMPARISON

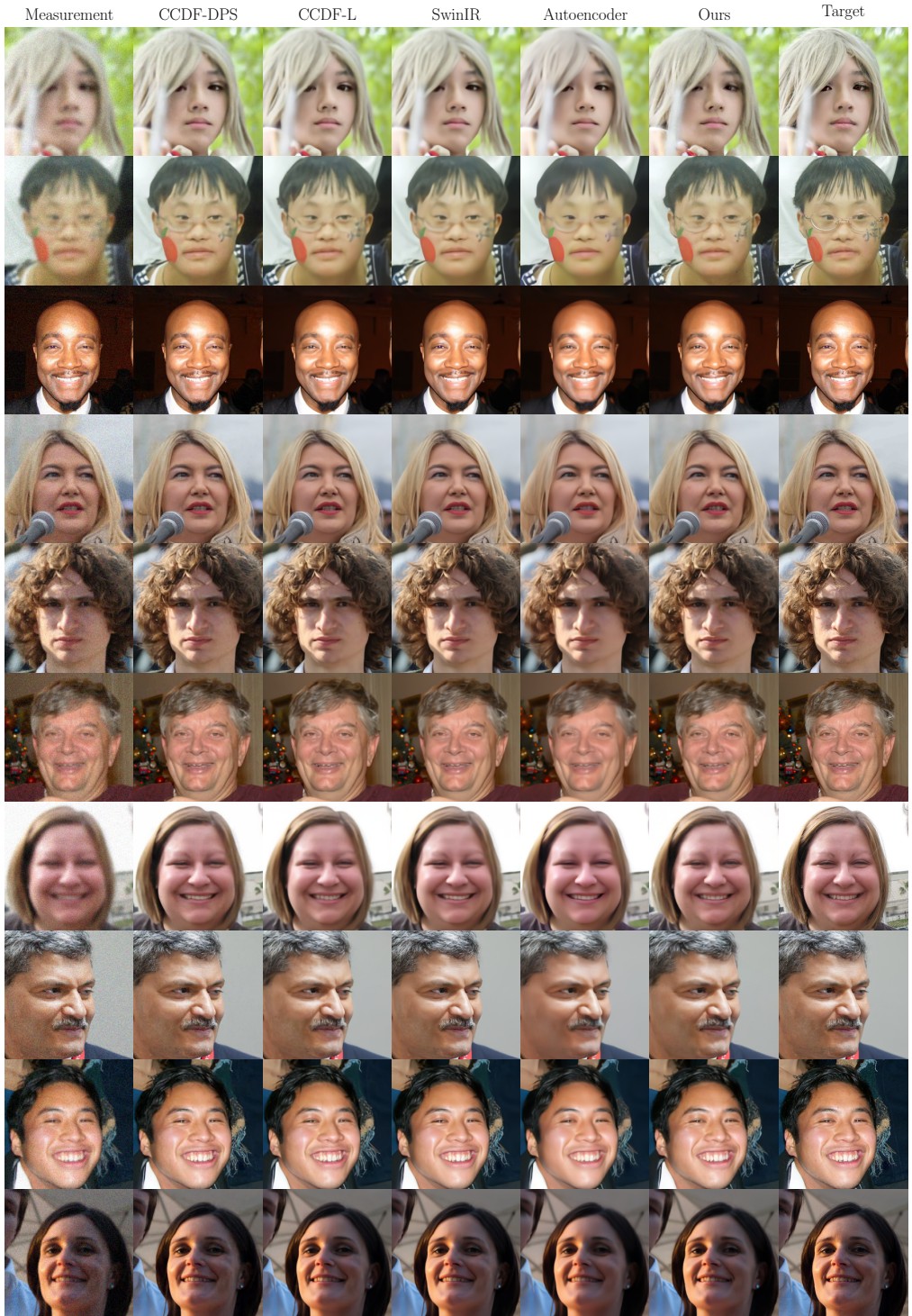

Figure 13: Reconstructed samples from the FFHQ test set on varying amounts of Gaussian blur with fixed noise $\sigma_{\boldsymbol{y}} = 0.05$.

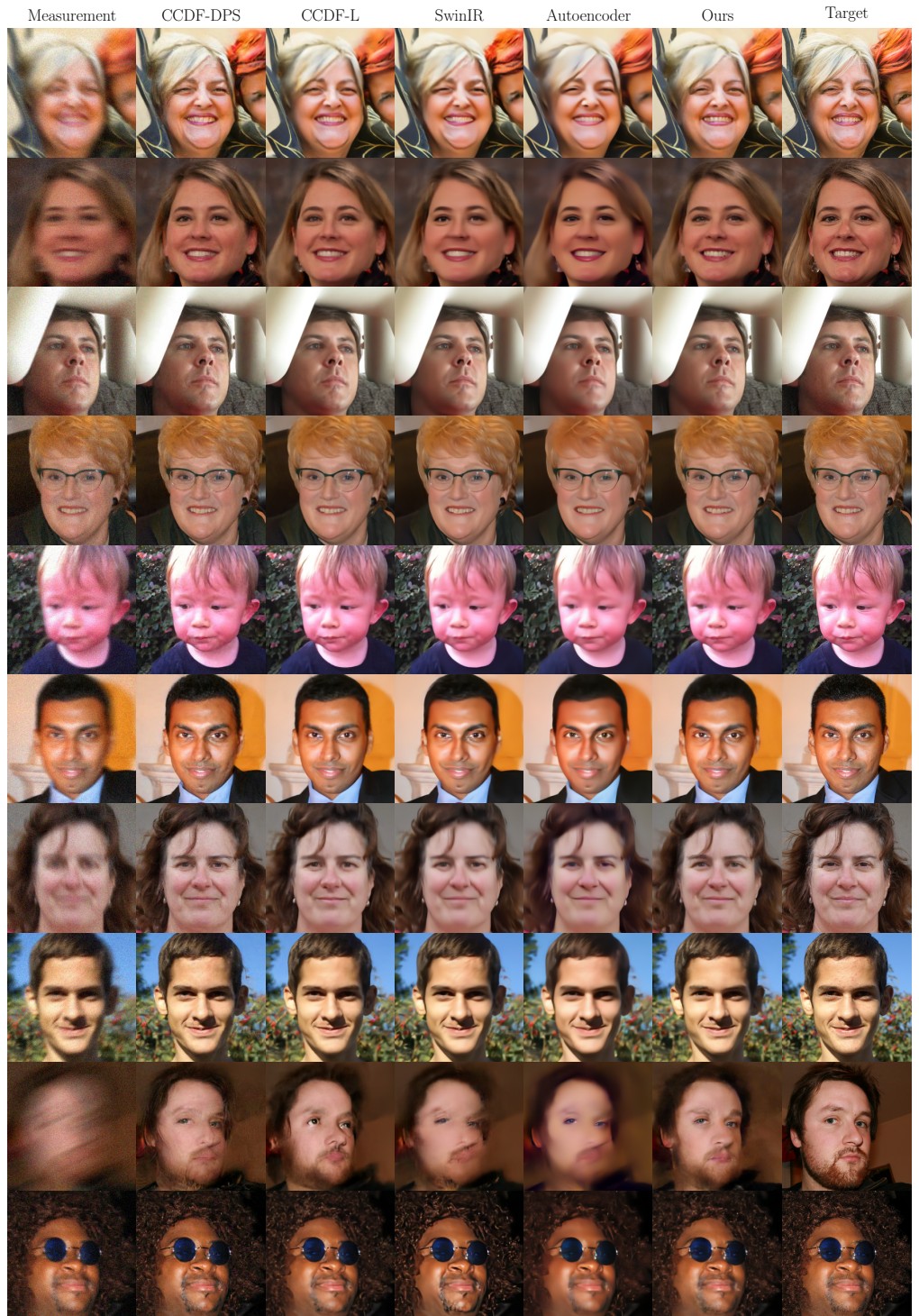

Figure 14: Reconstructed samples from the FFHQ test set on nonlinear motion blur with random kernels and fixed noise $\sigma_{\boldsymbol{y}} = 0.05$.

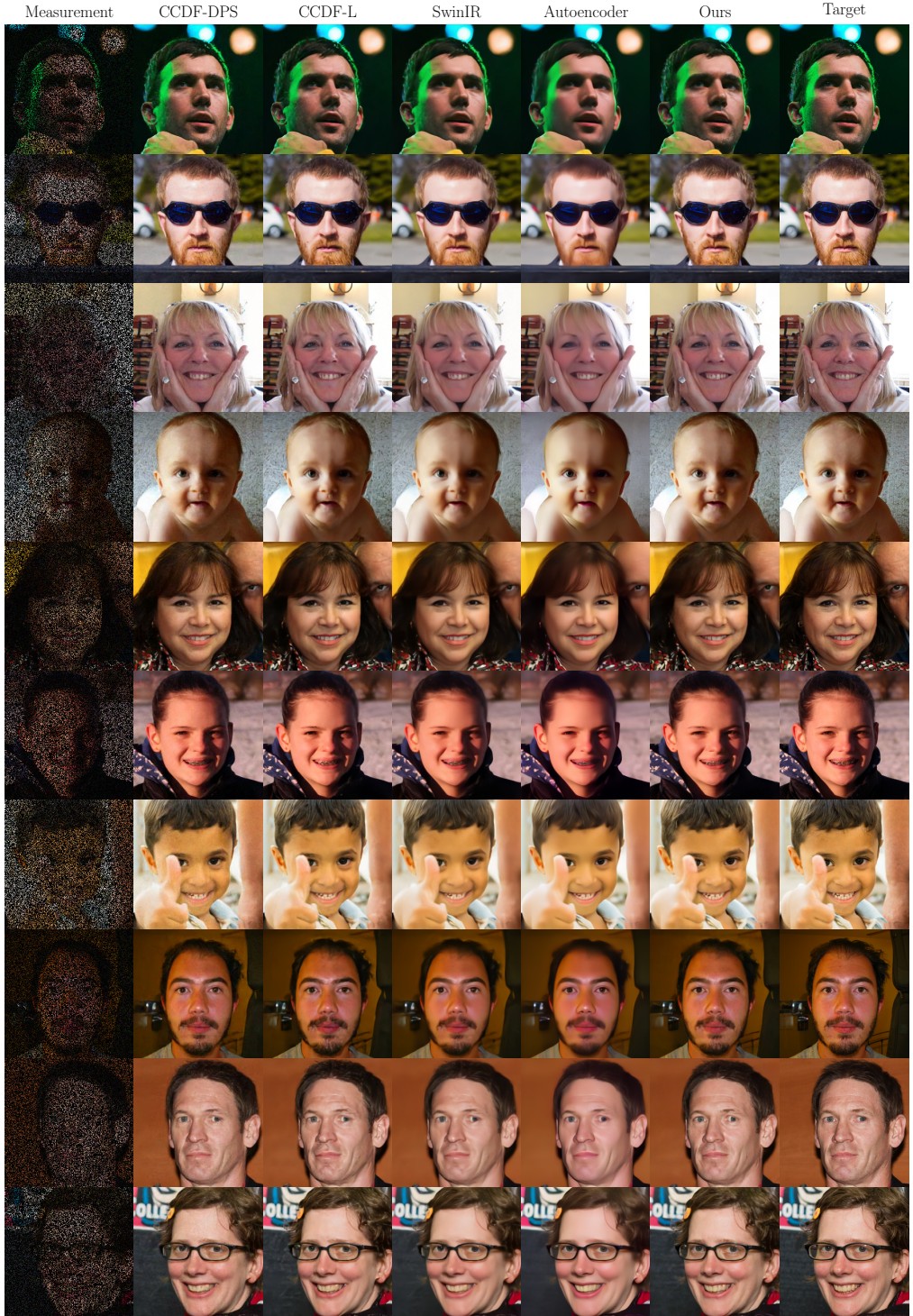

Figure 15: Reconstructed samples from the FFHQ test set on varying amounts of random inpainting with fixed noise $\sigma_{\boldsymbol{y}} = 0.05$.

