# OpenReview forum: "Adapt and Diffuse: Sample-adaptive Reconstruction via Latent Diffusion Models"
_ICLR.cc/2024/Conference — Submitted to ICLR 2024_

### Official Review · Reviewer_VtJz · 2023-10-29

**Soundness:** 2 fair
**Presentation:** 3 good
**Contribution:** 2 fair
**Rating:** 5
**Confidence:** 3

**Summary:**

This paper proposes a method to quantify the degradation severity of signal reconstruction in inverse problems using diffusion models. The key idea is to evaluate the severity in a latent space defined by variational autoencoder models. The severity is strongly correlated with the true corruption level and provides useful hints at the difficulty of signal reconstruction on a sample-by-sample basis. The usefulness of the severity measure is experimentally demonstrated.

**Strengths:**

The severity measure proposed in this article seems somewhat correlated with the difficulty of signal reconstruction. Computational experiments show that it leads to the improvement of reconstruction quality and the reduction of necessary computational cost.

**Weaknesses:**

There is little theoretical backing. Experimental results are not enough to support the validity of the proposed method. There are various hyper parameters to tune, which makes the method less attractive.

**Questions:**

How were hyper-parameters "\lambda_\sigma", "\lambda_im", "c", etc. tuned? Do we need to tune them for each data set? Or, they little vary depending on data sets?

---

> ### Author Response · Authors · 2023-11-18
> **Response to Reviewer VtJz**
>
> We thank the reviewer for the effort reviewing our paper. We are glad that the reviewer thinks that our experiments show that Flash-Diffusion “leads to the improvement of reconstruction quality and the reduction of necessary computational cost”. This is in fact the main focus of our paper. Please find below our response to your concerns.
>
> - Re theoretical backing: We position our work as a thorough empirical study of a concept that is highly underexplored in the modern inverse problem literature: sample-adaptivity. We hope that the reviewer can appreciate the novelty of the severity encoding framework and the clear demonstration of gains provided by Flash-Diffusion both in terms of efficiency and reconstruction performance. That said, it would be interesting to investigate some theoretical properties of the framework in future work and we would be open for suggestions on specific ideas the reviewer thinks can further strengthen our paper.
>
> - Re experimental results: We understand that the reviewer needs more experimental results in order to better gauge the validity of the results.  We have added the following new experiments to the manuscript:
>
>   1. A complete set of experiments on a new forward operator: random inpainting.
>
>   2. In-depth ablation studies against shifts in noise level and forward model (*Appendix D*).
>
>   3. An ablation study on the usefulness of various components of our framework (*Appendix C*).
>
>   4. A set of experiments on pairing severity encoding with DDIM sampling (*Appendix E*).
>
>   We hope that these additional experimental results convinced the reviewer on the validity of our method. We are open for further specific suggestions the reviewer would find necessary to incorporate into the paper.
>
> - Re hyperparameters: We detail how we tuned each hyperparameter in *Appendix A*. We would like to highlight that we have the same number of hyperparameters as other diffusion-based reconstruction methods, such as CCDF-DPS, and only one extra hyperparameter compared with DPS, the correction factor $c$. However, we find that similar values of $c$ work across all forward operators. The hyperparameter $\lambda_{\sigma}$ controls the relative importance of error estimation in the loss, and we tuned it for Gaussian blur over grid search ($[0.1, 1.0, 10]$) and used the same value for all other models with satisfactory results. That is, we find that no extensive hyperparameter tuning is required across tasks. Moreover, we simply scale the reconstruction loss terms $L_{im. rec.}$ and $L_{lat. rec.}$ with their corresponding dimension (pixel space dimension and latent space dimension respectively) without any further hyperparameter tuning across all experiments. Thus, we conclude that severity encoder training is fairly robust against these hyperparameters.
>
> We hope that we have addressed the reviewer’s concerns adequately and are happy to discuss further if there are any remaining concerns.

---

> > ### Comment · Reviewer_VtJz · 2023-11-21
> >
> > Thank you for the response. I appreciate the effort the authors made for showing the usefulness for various purposes. I raise my score slightly.

---

> > > ### Author Response · Authors · 2023-11-21
> > > **Response to Reviewer VtJz**
> > >
> > > Thank you for your response and for checking our updated manuscript and raising your score. Please let us know if you have any specific concerns left with respect to our work, we will do our best to address it further within the time frame of the rebuttal.

---

### Official Review · Reviewer_Gc37 · 2023-10-30

**Soundness:** 2 fair
**Presentation:** 2 fair
**Contribution:** 2 fair
**Rating:** 5
**Confidence:** 4

**Summary:**

In this work, to address the problem that most existing solvers cannot adapt their computing power to the difficulty of the reconstruction
task, the authors propose the severity encoding method, which estimates the degradation severity of noisy and degraded signals in the latent space of an autoencoder. Based on latent diffusion models and the severity encoding method, the authors propose the Flash-Diffusion method, which is a sample-adaptive method that fine-tunes the diffusion sampling trajectory based on the severity of corrupted signals. The authors claim to experimentally demonstrate that the proposed method achieves performance on par with SOTA diffusion-based reconstruction approaches, but with greatly improved compute efficiency.

**Strengths:**

The proposed severity encoding method may be of interest.

**Weaknesses:**

1. The proposed Flash-Diffusion method appears to be a straightforward blend of the severity encoding method and certain existing techniques/tricks that have been previously proposed in prior works, such as the technique proposed by Chung et al. (2022c) that starts the reverse diffusion process from a good initial reconstruction instead of pure noise.

2. The severity encoding method seems to be a minor trick. It is based on a restrictive and impractical assumption that the prediction error in latent space can be modeled as zero-mean i.i.d. Gaussian, despite the authors making some efforts to mitigate this assumption through noise correction.. While the presented experimental results may provide some insight into the potential effectiveness of this technique, I believe that they are insufficient to prove the viability of it.

3. The authors highlight in the abstract that "our technique achieves performance comparable to state-of-the-art diffusion-based techniques, with significant improvements in computational efficiency". But they only provide a very short paragraph before **Conclusions** to illustrate the efficiency of the method, and there is a clear lack of comprehensive comparisons or benchmarking against other techniques, particularly in terms of computational efficiency. Additionally, Figure 6, which is meant to illustrate the efficiency of the method, is confusingly labeled with an x-axis for "NFE" and a legend for "fixed steps", making it difficult to interpret. Furthermore, while the authors mainly try to achieve significant improvements in computational efficiency, they still stick to DDPM and use more than 100 NFEs. In terms of sampling efficiency, DDPM is rather weak and outdated. The authors should at least try to combine their method with DDIM and DPM-Solver, or other more advanced fast sampling methods for diffusion models (which only require less than 20 NFEs to obtain reasonable generation/reconstruction).

4. As far as I can tell, the authors only present experimental results for the deblurring task, but mention in the abstract that "We perform numerical experiments on both linear and nonlinear **inverse problems**". For "inverse problems", I would expect to see the results for various types of inverse problems, instead of only for deblurring. Please be more precise.

**Questions:**

See the weaknesses.

---

> ### Author Response · Authors · 2023-11-18
> **Response to Reviewer Gc37 (pt 1)**
>
> We appreciate the reviewer’s time reviewing our work. Please find below our response to your concerns.
>
> - Re Weakness 1: First, we would like to point out that the idea of adapting the cost of reconstruction to the severity of degradation has not been explored in detail to the best of our knowledge in the literature. Severity encoding is our novel contribution to fill this gap. We introduce severity encoding as a general framework that can be seamlessly combined with existing diffusion solvers to add adaptivity to such techniques. We don’t believe that this is a limitation of our work, but a property that adds extra flexibility to our reconstruction framework. Moreover, we have indeed discussed the connection of our work to *Chung et al. (2022c)* in *Sections 2* and *3* and demonstrated significant improvements over their technique in terms of reconstruction quality (see CCDF-DPS in *Table 1*).
>
> - Re Weakness 2: As the reviewer mentions, our severity estimates build upon the assumption that the estimation error in latent space is i.i.d. Gaussian. We agree with the reviewer that this assumption does not necessarily hold in practice. Indeed this is something we discuss in the paper and propose a technique to mitigate it. It would be an exciting direction for future work to investigate more nuanced error models for severity encoder training, such as modeling as Gaussian with general diagonal covariance. We however respectfully disagree about the impracticality of the assumption as our in-depth experiments demonstrate that the learned severity estimates (1) correlate well with the true underlying image degradation level (*Figures 3* and *4*), (2) can provide useful insight into outlier samples in the dataset in terms of reconstruction difficulty (*Figure 4*) and (3) can be effectively leveraged by diffusion-based solvers to accelerate sampling and improve reconstruction quality (*Table 1* and *Figure 6*). We are open for specific suggestions on how to further demonstrate the viability of our proposed framework.
>
> - Re Weakness 3:
>
>   - Re efficiency: We highlight the efficiency of our method by demonstrating that any non-adaptive/fixed starting diffusion time would result in worse performance than our sample-adaptive method. This key experiment is depicted in *Figure 6*, which we understand the reviewer found confusing. We updated the figure and its caption to more clearly reflect our efficiency results. To highlight our findings based on *Figure 6* and *Table 1*: we achieve better reconstruction quality using approx. 100 diffusion steps on average than DPS with 1000 diffusion steps. Moreover, using any fixed/non-adaptive starting time in the latent diffusion framework achieves worse performance than sample-adaptive Flash-Diffusion.
>
>   - Re using other sampling techniques: Thank you for suggesting the combination with DDIM, this is a great idea. Even though we are familiar with the efficiency of DDIM in image generation, we are unaware of general (including nonlinear) inverse problem solvers that leverage latent space DDIM and achieve performance comparable to state-of-the-art in reconstruction. We would appreciate it if the reviewer could point us to such techniques, we would gladly compare them with Flash-Diffusion. That said, we added two new sets of experiments to address the reviewer’s question on pairing severity encoding with DDIM.
>
>     1. We replace DDPM sampling (N=1000) with DDIM sampling (N=20 and N=100) in Flash-Diffusion in order to investigate the viability of such accelerated sampling schemes for latent space reconstruction. We observe that Flash-Diffusion obtains high quality reconstructions with as low as 10 NFEs, however there is a clear degradation in performance in terms of image quality metrics (see *Table 5*). We hypothesize that this can be due to the inaccuracy of LDPS in the accelerated sampling setting, which is an interesting direction for future work. Furthermore, we find that Flash-Diffusion reconstruction quality is robust in terms of the DDIM sampling stochasticity parameter. See a detailed discussion and experimental results in *Appendix E*.
>
>     2. We demonstrate that naively running DDIM sampling in latent domain with LDPS, even though greatly reduces sampling cost, is not competitive with other diffusion-based solvers.  In particular, we experiment with LDPS with DDIM sampling of N=100, which closely matches the average number of steps Flash-Diffusion uses in the DDPM setting. We observe that reconstruction quality has extreme variance: the technique produces reasonable reconstructions on some samples, but completely breaks down on others resulting in poor performance on average (see *Table 6*).  A comprehensive discussion is added to *Appendix E*.

---

> > ### Author Response · Authors · 2023-11-18
> > **Response to Reviewer Gc37 (pt 2)**
> >
> > - Re Weakness 4: Thank you for pointing this out. We added a new set of experiments on random inpainting in order to further demonstrate the viability of severity encoding across different inverse problems. We find that the gains of Flash-Diffusion in terms of performance and efficiency extend to random inpainting as well. We hope that these new experiments help convince the reviewer about the contribution of our work.
> >
> > We appreciate the feedback and hope that the additional results and clarifications have improved the reviewer’s opinion on our work. We are happy to discuss further if there are any other remaining issues.

---

> > > ### Author Response · Authors · 2023-11-21
> > > **Asking Reviewer Gc37 for feedback on the updated manuscript**
> > >
> > > We did our best to address the concerns raised by the reviewer and added experiments on a new forward operator and comparisons in the DDIM framework. We hope that the updated version of our manuscript can convince the reviewer about the merit of our work, and would greatly appreciate your feedback on the current version of the manuscript.

---

> > > > ### Comment · Reviewer_Gc37 · 2023-11-22
> > > > **Response to rebuttals**
> > > >
> > > > I thank the authors for the detailed responses and the added experimental results.
> > > >
> > > > However, I still think that it is strange for a paper pursuing efficiency to use DDPM, which leads to very weak baseline methods for reconstruction (in terms of efficiency). As far as I can tell, pursuing efficiency by using deterministic ODE samplers such as DDIM and DPM-Solver is a much more promising direction (which may have the potential to reduce the required NFE from 1000 to less than 20, although may require some additional engineering efforts; and I found that it is a bit strange that the authors cannot obtain some desired improvements in efficiency even for DDIM, which can be thought of as a simple deterministic adaption of DDPM (I want to clarify that for DDIM, 20 NFEs may not be sufficient; I mention 20 NFEs mainly for more advanced samplers such as DPM-Solver)). Along with the remained concern about the significance of the proposed severity encoding method, I would like to only raise my score slightly.

---

> > > > > ### Author Response · Authors · 2023-11-22
> > > > > **Response to Reviewer Gc37**
> > > > >
> > > > > Thank you for getting back to us and updating your rating. We do agree that combining severity encoding with deterministic ODE samplers is a promising direction for further accelerating conditional sampling. We believe that the greatest challenge in this direction is maintaining data consistency with the measurement in such samplers, due to the fact that the popular (L)DPS approximation can be inaccurate in this setting leading to degradation in performance. Finding a solution to this issue is a worthwhile goal in itself and a very interesting direction for future work, however it is out of the scope of this paper as our current focus is to demonstrate sample-adaptivity via severity encoding.

---

### Official Review · Reviewer_fcte · 2023-10-31

**Soundness:** 3 good
**Presentation:** 3 good
**Contribution:** 2 fair
**Rating:** 6
**Confidence:** 2

**Summary:**

This paper introduces a method called severity encoding that estimates degradation severity of noisy and degraded signals in the latent space of an autoencoder. The proposed reconstruction method, based on latent diffusion models, leverages the predicted degradation severities to fine-tune the reverse diffusion sampling trajectory and achieve sample-adaptive inference times. The technique aims to address the limitations of existing reconstruction techniques by adapting compute power to the difficulty of the reconstruction task. The authors demonstrate through numerical experiments on linear and nonlinear inverse problems that their approach achieves comparable performance to state-of-the-art diffusion-based techniques while significantly reducing computational cost. The paper also provides background information on diffusion models, denoising diffusion probabilistic models (DDPMs), and latent diffusion models (LDMs), as well as their applications in solving inverse problems by running diffusion in the latent space of a pre-trained autoencoder. Moreover, the proposed method shows promise in improving reconstruction efficiency while maintaining performance.

**Strengths:**

Originality: The paper introduces severity encoding to estimate degradation severity in an autoencoder's latent space, providing a fresh perspective on reconstruction challenges. Combining severity encoding with latent diffusion models sets it apart.

Quality: Thorough experiments on linear and nonlinear inverse problems demonstrate effectiveness. Comparisons with state-of-the-art diffusion-based methods show similar performance with reduced computational costs. Well-designed experiments and comprehensive analysis ensure reliable findings.

Clarity: The paper effectively communicates the proposed method and its technical details.

Significance: The contributions have implications for image restoration and inverse problems. Severity encoding allows sample-adaptive inference, addressing a major limitation of existing methods. This has the potential to improve efficiency and effectiveness in various domains by reducing computational costs while maintaining performance.

**Weaknesses:**

Limited discussion on potential limitations: The paper does not thoroughly address potential limitations and challenges associated with severity encoding. It is crucial to identify and explicitly discuss any limitations, such as the impact of inaccurate severity predictions on overall reconstruction performance and scenarios where severity encoding may struggle to provide accurate estimates. This would provide a nuanced understanding of the method's capabilities and constraints.

Lack of ablation studies: The paper lacks ablation studies to assess the individual impact of different components or design choices in the proposed method. Conducting ablation experiments to investigate the contributions of severity encoding, latent diffusion models, and other key components would help understand their relative importance and guide further improvements or adjustments to the approach.

**Questions:**

1. Regarding the limited discussion on potential limitations and challenges associated with severity encoding:
   - Could the authors elaborate on potential scenarios or data conditions where severity encoding may lead to inaccurate severity predictions and the resulting impact on overall reconstruction performance?
   - Are there any strategies or techniques that can be employed to mitigate the limitations of severity encoding in scenarios where it may struggle to provide accurate estimates?

2. Concerning the lack of ablation studies to assess the individual impact of different components in the proposed method:
   - Could the authors provide insights into the relative importance of severity encoding, latent diffusion models, and other key components within the proposed approach based on their expertise and experimentation?
   - Are there specific aspects of the method that could benefit from further refinement or adjustments based on the results of potential ablation experiments?

---

> ### Author Response · Authors · 2023-11-18
> **Response to Reviewer fcte**
>
> Thank you for the insightful review. We are glad that the reviewer agrees that our work on sample-adaptive inference “addresses a major limitation of existing methods” and that our experiments are “well-designed” and the analysis is “comprehensive”. Please find below our response to your concerns.
>
> - Re discussion on potential limitations: Thank you for pointing this out along with **Reviewer Cd1z**. We agree that understanding the limitations of severity encoding is crucial. Therefore, we added an in-depth robustness study investigating the effects of noise level perturbations and mismatch in forward model on severity encoding. To summarize our findings:
>
> 1. **Noise perturbations in test time:** Interestingly, we find that severity encoding maintains its performance at noise levels lower than in the training setting (*Figures 8* and *9*), which results in steady reconstruction performance of Flash-Diffusion. At higher unseen noise levels severity encoding accuracy decreases, however the overall degradation in Flash-Diffusion performance is more graceful than in case of other supervised  baselines (*Figure 9*). We hypothesize that this effect is due to adaptivity: the severity estimates are still useful in finding an appropriate starting time, however not as accurate as at familiar noise levels. Moreover, we observe that the severity encoder manages to scale the compute effort at noise levels unseen during training: for measurement noise variances outside the training setting, the predicted number of necessary diffusion steps is proportional to the noise level (*Figure 10*).  However, at extreme noise perturbations, the severity encoder completely breaks down, assigning essentially the same severity estimate to any input.
>
> 2. **Forward model perturbations in test time**: we observe a consistent degradation in performance when the forward model is drastically changed compared to the training setup (e.g. training on Gaussian blur and evaluation on nonlinear blur, *Table 4*). However, we find that as long as the operators are somewhat similar, the drop in performance is more modest. For instance, we show that a severity encoder trained on non-linear blur has high accuracy on estimating the degradation severity of images corrupted by Gaussian blur, *as long as the blur amount is high* (*Table 3* and *Figure 11*). This observation may hint at the potential for generalizing to unseen operators when the severity encoder is trained on a mix of known operators, which is an exciting direction for future work.
> We have also highlighted these limitations in the main text in *Section 4* and we are planning to incorporate the above experiments into the main manuscript in the camera-ready version (given one extra page). With respect to strategies to mitigate severity encoder failure: this is a very interesting question. As we observe in the robustness experiments, the severity encoder remains accurate at *lower* unseen noise levels. Therefore, a technique to improve noise robustness would be to train the encoder on slightly higher noise levels than expected in test time in order to “proof” it against potential failure when the noise level is perturbed.
>
> - Re ablation studies: This is a great point. We added a new set of experiments in *Appendix C*, where we ablate the effect of the various components of Flash-Diffusion as the reviewer suggested. As seen in *Table 2*, adaptivity via severity encoding has the largest contribution to performance.  We obtain comparable results with and without latent space diffusion, however latent diffusion is preferable since (1) it has greatly reduced computational cost and (2) degradation severity is better captured in latent space. We hope the reviewer finds this new ablation study useful.
>
> We are happy to discuss any further questions or concerns and would like to ask the reviewer to consider raising their score in case we addressed all major issues.

---

> > ### Author Response · Authors · 2023-11-21
> > **Asking Reviewer fcte for feedback on the updated manuscript**
> >
> > As the end of the rebuttal period is approaching, we would like to ask the reviewer for their feedback on our updated manuscript. We did our best to address all concerns and we believe that our work became stronger after incorporating all your feedback. We would greatly appreciate it if the reviewer could let us know about any further concerns and update their scores reflecting their opinion on the current version of the work.

---

> > ### Comment · Reviewer_fcte · 2023-11-22
> >
> > Thanks for the response. I do not have further questions.

---

### Official Review · Reviewer_Cd1z · 2023-11-06

**Soundness:** 3 good
**Presentation:** 4 excellent
**Contribution:** 3 good
**Rating:** 8
**Confidence:** 3

**Summary:**

This paper proposes to adapt the amount of computation in image restoration based on posterior sampling with a latent diffusion model, by taking into account the sample-specific "severity" of degradation. This severity is estimated using the proposed "severity encoder". Experiments show that the proposed strategy outperforms comparable models in terms of speed while retaining (or improving) quality.

**Strengths:**

- The paper is very well written and easy to follow. The main idea of the paper is explained well and supported by experiments: samples with significant degradation require more steps for clean reconstruction, while less degraded samples need fewer diffusion steps. Handling this properly can prevent over-diffusion and save computational resources.

- The idea of sample-adaptive computation is natural in seems to be new in this context. I think that this is implicit in more traditional solvers (e.g. compressed sensing) but certainly not for neural nets.

- As Figure 3 shows, the auto-encoder-based “severity” measure aligns well with objectiv and perceptual severity.

**Weaknesses:**

- while diffusion models can serve as a prior for solving inverse problems in an unsupervised manner [1], the proposed method relies on supervised learning. It thus needs paired to train the severity encoder. One consequence, as shown in Appendix C, is that the reconstruction quality can substantially deteriorate for measurements not used during  training. The main text lacks a clear acknowledgment of this important drawback. It's important to incorporate (parts of) Appendix C in Section 4 and discuss this more explicitly in the main text.

- There is no discussion of alternative heuristics to measure severity of degradation

- It would be nice to evaluate the robustness of the proposed framework to variations in measurement noise levels in test time, as these changes are likely to happen in practice

- It would be nice to analyze the estimated degradation severity in both the above experiment and the ones conducted in Appendix C. This could ascertain the robustness of the proposed sample-by-sample computation adaptation.


[1] Hyungjin Chung, Jeongsol Kim, Michael T Mccann, Marc L Klasky, and Jong Chul Ye. Diffusion posterior sampling for general noisy inverse problems. arXiv preprint arXiv:2209.14687, 2022a.

**Questions:**

- since that the original formulation of diffusion generative models is not computationally efficient, a number of papers propose streamlined versions [2,3]. Their “diffusion” methods don’t require the standard iterations. Could you discuss the potential of your ideas in this alternative context?


[2] Song, Y., Dhariwal, P., Chen, M. and Sutskever, I., 2023. Consistency models.

[3] Shao, S., Dai, X., Yin, S., Li, L., Chen, H. and Hu, Y., 2023. Catch-Up Distillation: You Only Need to Train Once for Accelerating Sampling. arXiv preprint arXiv:2305.10769.Vancouver

---

> ### Author Response · Authors · 2023-11-18
> **Response to Reviewer Cd1z**
>
> We are grateful for the helpful feedback. We are glad that the reviewer thinks that the “main idea of the paper is explained well and supported by experiments” and that the proposed sample-adaptive computation is natural and novel. Please find below our response to the issues the reviewer has raised.
>
> - Re acknowledgment of limitations and robustness studies (bullets 1 and 3): Thank you for raising this point. We agree that it is crucial to understand the limitations of the severity encoder, especially in the face of realistic test-time perturbations, including noise level mismatch and forward model shifts. To address your point, we have performed a new set of robustness studies that dive deeper into this question. To summarize our findings:
>
> 1. **Noise perturbations in test time:** Interestingly, we find that severity encoding maintains its performance at noise levels lower than in the training setting (*Figures 8* and *9*), which results in steady reconstruction performance of Flash-Diffusion. At higher unseen noise levels severity encoding accuracy decreases, however the overall degradation in Flash-Diffusion performance is more graceful than in case of other supervised  baselines (*Figure 9*). We hypothesize that this effect is due to adaptivity: the severity estimates are still useful in finding an appropriate starting time, however not as accurate as at familiar noise levels. Moreover, we observe that the severity encoder manages to scale the compute effort at noise levels unseen during training: for measurement noise variances outside the training setting, the predicted number of necessary diffusion steps is proportional to the noise level (*Figure 10*).  However, at extreme noise perturbations, the severity encoder completely breaks down, assigning essentially the same severity estimate to any input.
>
> 2. **Forward model perturbations in test time**: we observe a consistent degradation in performance when the forward model is drastically changed compared to the training setup (e.g. training on Gaussian blur and evaluation on nonlinear blur, *Table 4*). However, we find that as long as the operators are somewhat similar, the drop in performance is more modest. For instance, we show that a severity encoder trained on nonlinear blur has high accuracy on estimating the degradation severity of images corrupted by Gaussian blur, *as long as the blur amount is high* (*Table 3* and *Figure 11*). This observation may hint at the potential for generalizing to unseen operators when the severity encoder is trained on a mix of known operators, which is an exciting direction for future work.
> We also agree that these limitations should be better highlighted in the main manuscript. Therefore, we added a discussion to *Section 4* on the above limitations and we plan to move the robustness studies to the main paper given an extra page in the camera-ready version.
>
> - Re other heuristics for degradation severity (bullet 2): We have a discussion on the challenges of quantifying degradation severity in pixel space in *Section 3*. That said, it would be interesting to compare the estimates from severity encoding with a heuristic baseline. We are open for suggestions in this direction for a simple and natural baseline.
>
> - Re bullet 3 on robustness to measurement variations: See discussion above where we simultaneously address bullets 1 & 3.
>
> - Re consistency models (bullet 4): Thank you for the references. Combining severity encoding with consistency models is indeed an interesting direction that can potentially result in few-shot high-fidelity reconstructions. Our adaptive shortcut idea works as long as SNR can be analytically evaluated for each time step. The crucial challenge would be that of maintaining data consistency, which has not been thoroughly investigated in the context of consistency models to our knowledge. This is a great suggestion however for future investigation.
>
> Please let us know if you have any further questions or suggestions. We hope that we managed to address all major concerns, in which case we would really appreciate it if the reviewer could update their rating given the generally favorable review.

---

> > ### Comment · Reviewer_Cd1z · 2023-11-21
> >
> > This authors have rather constructively addressed my concerns so I'll raise my score.

---

> > > ### Author Response · Authors · 2023-11-21
> > > **Response to Reviewer Cd1z**
> > >
> > > Thank you for engaging constructively with our work and we appreciate you updating your score.

---

### Author Response · Authors · 2023-11-18
**Summary of updates to the manuscript**

We are grateful for all the insightful comments and feedback from the reviewers. We did our best to address all the concerns that the reviewers have raised and updated the manuscript accordingly. An overview of the updates is as follows:

- We have added a new set of experiments on random inpainting in order to address **Reviewer Gc37**’s point wanting to see more forward models in addition to Gaussian blur and nonlinear motion blur. This can be found in *Table 1* in the main portion of the manuscript.

- Following the suggestion of **Reviewer Cd1z** and **Reviewer fcte**, we added in-depth ablation studies on the robustness of both severity encoding and Flash-Diffusion in order to further improve our understanding of the limitations of the technique in the face of test-time perturbations in measurement noise level and forward operator. These experiments can be found in *Appendix D*.

- Addressing the call of **Reviewer fcte** for experiments ablating different components of our framework, we added new ablation studies analyzing how each component contributes to the performance of Flash-Diffusion. Details are included in *Appendix C*.
Following **Reviewer Gc37**’s suggestion to investigate other samplers, we added a new set of experiments showcasing (1) the performance of Flash-Diffusion when paired with DDIM sampling and (2) a comparison with a naive latent domain DDIM baseline. For details, see *Appendix E*.

- We updated *Figure 6* to clarify the depicted setting, as it was found to be confusing by **Reviewer Gc37**.

- We added extra reconstruction samples for visual inspection in *Appendix H*.

We have highlighted the new additions in the manuscript in red. We believe that the above updates further strengthened our work and we hope that we were able to address the reviewers’ concerns. That said, we are happy to address any further questions or concerns and look forward to active discussions during the rebuttal period.

---

### Meta-Review · Area_Chair_hUFs · 2023-12-16

**Metareview:**

This work proposed an approach to dynamically adjust the computational workload for image restoration. This adjustment is achieved through posterior sampling utilizing a latent diffusion model, and it considers the individual "severity" of degradation associated with each sample. To estimate this severity, the work introduces a new "severity encoder."  Experiments demonstrate that this strategy surpasses similar models in terms of processing speed while maintaining or even improving the quality of the restored images.

This work is well-written and well-presented, which addresses an important efficiency issue for solving inverse problems via diffusion models. However, the reviewers are concerned about the practicality of the approach, due to (1) extra hyperparameters for tuning, (2) can only be applied to DDPM which is not SOTA sampling method, and (3) lack of comprehensive experiments and theoretical support.

We encourage the authors to incorporate the reviewers' comments for future submission.

**Justification For Why Not Higher Score:**

There are several major concerns of the paper that cannot be addressed via rebuttal, such as the extension to DPM and DDIM sampler, limiting the practicality and impact of the approach

**Justification For Why Not Lower Score:**

n/a

---

### Decision · Program_Chairs · 2024-01-16

Reject